# Responses of riverine dissolved organic matter to damming in two distinct hydrological regimes of Northern Spain

Selin Kubilay[1, 2, 3], Edurne Estévez[2], José Barquín Ortiz[4], and Gabriel Singer[2]

[1]Department of Ecohydrology and Biogeochemistry, Leibniz Institute of Freshwater Ecology and Inland Fisheries, Berlin, 12587, Germany
[2]Department of Ecology, University of Innsbruck, Innsbruck, 6020, Austria
[3]Geography Department, Humboldt University of Berlin, Berlin, 12489, Germany
[4]IHCantabria - Instituto de Hidráulica Ambiental de la Universidad de Cantabria, Santander, 39011, Spain

**Correspondence:** Selin Kubilay (kubilay.selin@gmail.com) and Gabriel Singer (gabriel.singer@uibk.ac.at)

**Abstract.** Iberian rivers are characterized by flow regimes with high seasonal flow variation. They also host one-fifth of Europe's reservoirs for hydropower generation, irrigation or water supply needs, and thus many rivers of this region have heavily altered flow regimes. Such flow conditions also alter the natural dynamics of dissolved organic matter (DOM), which likely has implications for carbon cycling due to changed conditions for the transformation, transportation, production, and storage of carbon. Here we looked into the effects of flow alteration on the "DOM regime", i.e. the seasonal variation of DOM concentration and composition, in 20 rivers belonging to two different natural (reference) flow regimes (i.e., Mediterranean and Atlantic) in Northern Spain. To further investigate which flow regime components influence DOM properties, we linked the observed seasonal shifts in DOM composition to a range of hydrological indices. We found that Atlantic rivers with a natural flow regime tended to have lower annual mean DOC concentration than their altered equivalents, this flow alteration trend is weakly mirrored in Mediterranean rivers. We did not observe much difference in annual average DOM composition due to flow alteration in either Atlantic or Mediterranean rivers. However, the seasonal variation in DOM composition was lower in altered Atlantic rivers compared to natural ones. This flow alteration effect on the DOM regime was not as distinctive in Mediterranean rivers, which showed a higher diversity of DOM regimes across rivers. We linked the lack of seasonal variation in DOM composition in flow-altered rivers mainly to the prevention of transmission of upstream-sourced DOM by the reservoirs. It appears that in our study area, reservoirs mostly act as a temporally homogenizing buffer averaging out naturally occurring shifts in DOM composition by transiently storing upstream-sourced carbon inputs and objecting them to bio and photo-degradation, thus sending relatively invariable DOM further downstream. This effect of dams on DOM regimes appears robust across both Atlantic and Mediterranean regimes, despite some heterogeneity of dam types and purposes, with potentially important consequences for riverine carbon cycling.

## 1 Introduction

The structure and functioning of a river ecosystem are strongly tied to its flow regime (Poff et al., 2006). Climate and weather interacting with catchment properties such as topography, geology and land cover are primary controls of natural flow dy-

namics (Poff and Zimmerman, 2010), yet natural flows are altered globally by a range of human interventions, including the construction and operation of dams for irrigation, navigation and hydropower generation (Stewardson et al., 2017). Traditionally, flow regimes are described by five components; magnitude, frequency, duration, timing and rate of change; and an altered flow regime is identified by deviation of one or more of these components from its natural tendency (Poff et al., 1997). Given the natural diversity of flow regimes and the many ways in which they can be modified by dam operations (McManamay et al., 2012; Nadon et al., 2015), the prediction of ecological consequences of flow alterations is not a trivial task.

Dissolved organic matter (DOM) is the largest pool of organic carbon and the main energy source for bacteria in aquatic environments (Battin et al., 2008; Riedel et al., 2012). DOM is chemically highly complex because of (i) the numerous sources it originates from, and (ii) the various biological and chemical processes it is subjected to during transport along the riverine network (Jaffé et al., 2008). Most of the DOM is of terrestrial origin, characterized by high aromaticity associated with humic acids (Jaffé et al., 2008), while only a small fraction is composed of highly reactive substances such as carbohydrates and proteins, which may have terrestrial as well as in-stream origin. This highly reactive fraction is often microbially degraded within minutes to hours after creation or entering the system while the more recalcitrant terrestrial fraction is transported along the river for a longer time (Hansen et al., 2016; Kaplan et al., 2008). This linkage of DOM reactivity with residence time creates a compositional DOM continuum unfolding over space and time (Catalán et al., 2016; Peter et al., 2020).

Spatiotemporal variation of DOM composition must also be tied to the flow regime (Hayes et al., 2018). In natural rivers, the variability of DOM concentration and composition is linked to (i) activation of heterogeneous flow paths draining water from the terrestrial surrounding to the river channel, especially during intense hydrologic events induced by snowmelt or rain (Raymond et al., 2016; Raymond and Saiers, 2010), and (ii) residence time of the DOM within the riverine network that is determined by discharge (Peter et al., 2020). High discharge spates may be "hot moments" for the mobilization and transport of terrestrial DOM (McClain et al., 2003), yet the decreased residence times in the river corridor may limit its microbial transformation. In contrast, during base-flow conditions, such terrestrial material is more dependent on soil processes and subsurface flow paths. Moreover, following high-flow events, discharge and concentration (or composition) of terrestrial DOM undergo uncoupling due to the limited terrestrial storage capacity and exhaustion of DOM reservoirs along drainage flow paths (Andrea et al., 2006; Wagner et al., 2019). Such hysteresis in discharge-concentration relationships combined with dynamic residence times prevents simple translations of flow regimes into "DOM regimes" (Andrea et al., 2006).

When flow dynamics are controlled by dams, flood extremes are minimized (Hayes et al., 2018) and water residence time increases due to the storage in the reservoirs, providing an opportunity to change the composition of DOM and altering the timing of its delivery to the downstream river (Xenopoulos et al., 2021). Recent studies indicate that DOM concentration and composition vary according to dam operation and purpose, also determining whether a reservoir acts as a source or sink of carbon (Chen et al., 2016; Kraus et al., 2011). A dam with sufficient residence time may dampen event-driven and seasonal terrestrial (allochthonous) signatures of DOM for downstream environments as the increased residence time in the reservoir enables photodegradation and biodegradation and may shift DOM composition towards a more autochthonous and more bi-olabile composition (Xenopoulos et al., 2021). The net effect of production, transformation and loss of organic carbon in a reservoir (as in any river reach) depends on several variables including the amount and composition of inflowing carbon, algal

and bacterial activity, nutrient availability, temperature and solar radiation (Kraus et al., 2011). Most of these seasonal biotic and abiotic factors are also associated with the river's natural flow regime; suggesting that the alteration of a specific flow regime may result in quite a specific DOM regime for the downstream river sections. Notably, temporal changes in DOM composition throughout the year, i.e. shifts in DOM composition (especially under the influence of flow alteration), have yet to be included in concepts of riverine carbon cycling.

Here, we study the effects of anthropogenic flow regime alterations by dams (which serve mainly hydropower and irrigation purposes) in rivers with two distinct natural (reference) flow regimes; namely, Atlantic and Mediterranean. Rivers were grouped into one of these two climate zone-associated flow regime types depending on either observed flow conditions (for natural, unaltered rivers) or given a classification into these reference regimes based on climate and landscape covariates (for those rivers altered by dams). Our objectives were to identify commonalities and differences in response to dam-induced flow alteration across Atlantic and Mediterranean rivers with regard to (i) mean and (ii) seasonally induced variation of DOM concentration and composition over one year as descriptors of a river´s DOM regime, and (iii) to assess spatial similarity of flow alteration effects on DOM regimes across multiple rivers (Fig. 1). We hypothesize that DOM regimes are shaped by natural flow regimes, but also reflect anthropogenic flow alteration. The natural reference flow regime expected for a river may constrain how a dam can impact the DOM regime. Alternatively, a dam could force a DOM regime irrespective of the underlying natural hydrology of a catchment, perhaps with more or less dependence on specific dam and reservoir features. Given the high spatial and temporal variance of DOM composition, determining commonalities in the response of DOM regimes to dam-induced flow alteration could increase our understanding of the ecological consequences of a dam on the downstream river´s structure and functioning, and thus improve our capability to predict effects on riverine carbon cycling.

## 2 Methods

### 2.1 Study design and sampling strategy

Our study followed a control-impact design comparing rivers with natural and altered flow regimes in two natural (reference) flow regime types in northern Spain (Fig. 1, Fig. 2). Peñas and Barquín (2019) classified hydrologically unimpacted Spanish rivers into 20 hydrological classes using data from 282 gauged naturally flowing rivers and relevant environmental and hydrological drivers such as climate, topography, land cover and geology. Using these latter discriminators, they defined natural equivalent flow regimes (in the sense of estimated pre-impact reference flow regimes) for rivers that are nowadays impacted by various types of flow alteration (Peñas et al., 2016). For our study, we selected 20 rivers within classes 13 and 10 of Peñas and Barquín (2019); in each class, we sampled 4 naturally flowing rivers and 6 rivers with altered flow regimes (where one altered Mediterranean river had to be removed). Sampling sites were distributed within six basins, but we excluded locations on mainstems and - except one site - no site´s catchment was nested within another one to maintain spatial independence (Fig. 2). All sampling locations were $2^{nd}$ to $4^{th}$ order rivers (Supp. Table. S1). For simplicity, the two (reference) flow regime types will be referred to as Atlantic and Mediterranean, respectively, since class 13 is located in the Atlantic-type climatic region and class 10 in the Mediterranean-type climatic region of Spain, regardless of where these rivers discharge or which basin they

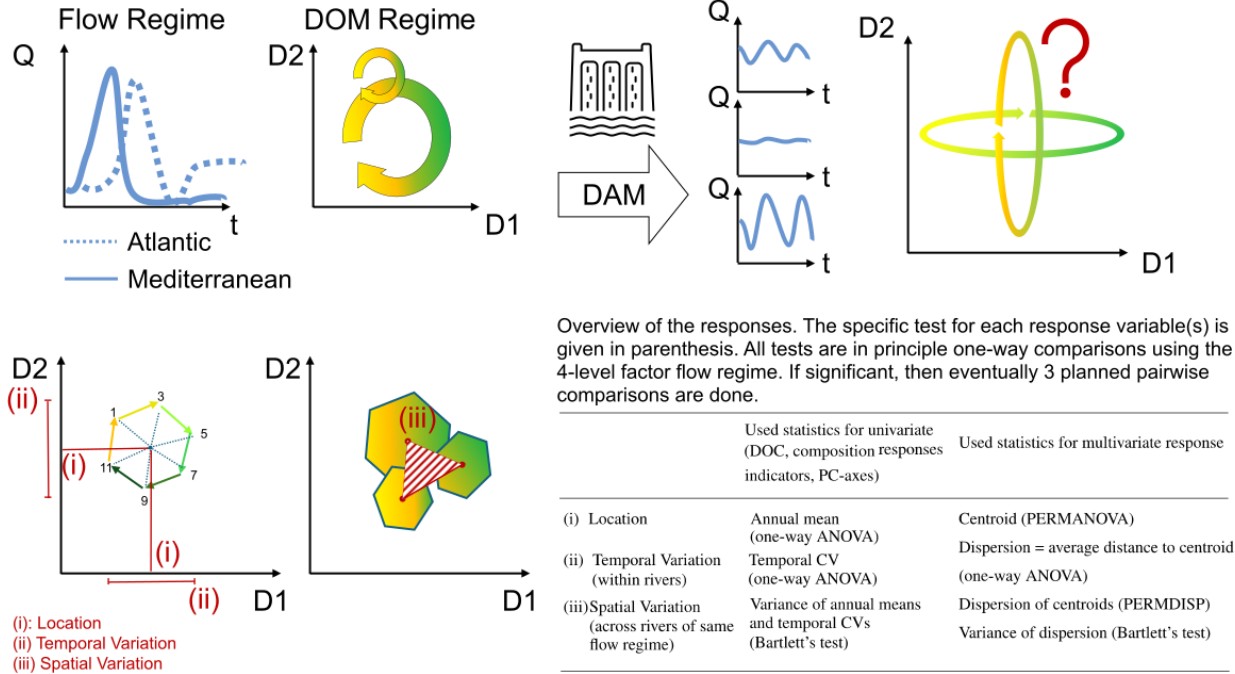

**Figure 1.** Conceptual summary of the objectives and statistical pipeline. The upper row of graph panels shows a river's hydrograph (shown as discharge Q over time t) translating into compositional shifts of DOM (shown as a circular shift in two-dimensional space that multiple descriptors of DOM chemistry could define). With the effect of a reservoir on a flow regime resulting in variously altered hydrographs, the consequences on the DOM regime remain unknown. The lower row illustrates how we investigated effects on DOM regimes by looking at the quality and quantity of DOM from 3 different angles; (i) the annual mean DOM composition (location), (ii) the temporal variation at an annual time scale, and (iii) the spatial variation of various DOM regime descriptors across several rivers. Our choice of statistical method depended on whether a univariate or a multivariate response was used.

belong to (Table 1, Fig. 2). Overall, in our study, the sampled sites, - referred to as rivers for simplicity - belong to one of four flow regimes: natural Atlantic (nA), altered Atlantic (aA), natural Mediterranean (nM) and altered Mediterranean (aM).

A natural Atlantic regime is characterized by highest flows in early spring (February-April) due to snowmelt, followed by a gradual decrease in flow until summer low flows are reached and continue throughout July-September (Fig. 3a). Once minimum flows are reached in October, the flow rate picks up quite quickly with a steep incline due to heavy rains in the fall. Contrary to this, a natural Mediterranean regime (Fig. 3d) has its highest flows a bit later in the spring, in March/April, followed by a sharp decrease in flow in May consequently leading to a long dry summer, reaching the minimum flows between August-September. As opposed to Atlantic rivers, the flow rate increases slowly leading back to the spring peaks, implying comparably less intense fall rains.

95

**Table 1.** Study design and environmental characteristics of the altered (a) and natural (n) Atlantic (A) and Mediterranean (M) flow regimes. The altered rivers encompass dams of various purposes/operations with varying reservoir volume(s). Sampling sites were located at similar distances to the next upstream reservoir, yet some sites had multiple upstream reservoirs. The catchment of Pisuerga is nested within the catchment of Aguilar, all other sites have spatially independent, non-nested catchments. Storage indices were taken from (Pompeu et al., 2022) and show storage volume/average annual runoff. The last column reports the ecological status of the next upstream-located reservoir according to the wfd (2021) database.

| River | Hydro Class | Alteration | Reservoir Usage | Upstream Catchment Area (km$^2$) | Elevation (m) | Reservoir (hm$^3$) | Storage Index | Distance to reservoir(s) (km) | Ecological Status |
|---|---|---|---|---|---|---|---|---|---|
| Aguilar | M | a | Irrigation | 579.87 | 882 | 247 - 65 | 0.82 | 5.5 - 33.6 | >Good |
| Pisuerga | M | a | Irrigation | 255.53 | 1014 | 65 | 0.46 | 2.2 | >Good |
| Carrion | M | a | Irrigation | 318.34 | 744.96 | 95 - 70 | 0.72 | 2.8 - 12 | >Good |
| Duero | M | a | Irrigation | 590.31 | 1032 | 249 | 0.74 | 8.3 | >Good |
| Ebro | M | a | Irrigation | 473.66 | 802 | 540 | 3.71 | 5.8 | Moderate |
| Luna | M | a | Irrigation | 507.69 | 1008 | 308 | 0.84 | 5.2 | >Good |
| Arlanzon | A | a | Irrigation | 173.24 | 1023 | 75 - 23.2 | 0.63 | 2.3 - 9.7 | >Good |
| Esla | A | a | Irrigation | 612.04 | 1001 | 664 | 0.86 | 3.1 | >Good |
| Nalon | A | a | Hydropower | 356.26 | 317.33 | 4-34 | 0.13 | 4.2 - 9.5 | >Good |
| Nansa | A | a | Hydropower | 363.37 | 66 | 2-12 | 0.06 | 1.74 - 29.1 | Good* |
| Narcea | A | a | Hydropower | 1258.45 | 122.33 | 33 | 0.03 | 4.4 | Moderate |
| Porma | A | a | Irrigation | 296.46 | 986 | 318 | 1.64 | 4.7 | >Good |
| Cea | M | n | - | 370.23 | 893.67 | - | | - | |
| Ega | M | n | - | 455.78 | 489 | - | | - | |
| Omana | M | n | - | 403.74 | 974 | - | | - | |
| Tiron | M | n | - | 186.59 | 815 | - | | - | |
| Bernesga | A | n | - | 88.95 | 1132 | - | | - | |
| Curueno | A | n | - | 168.29 | 1035.33 | - | | - | |
| Deva | A | n | - | 647.17 | 23.68 | - | | - | |
| Sella | A | n | - | 476.28 | 60 | - | | - | |

*We could not find reservoir information, instead checked upstream and downstream water bodies for their ecological status*

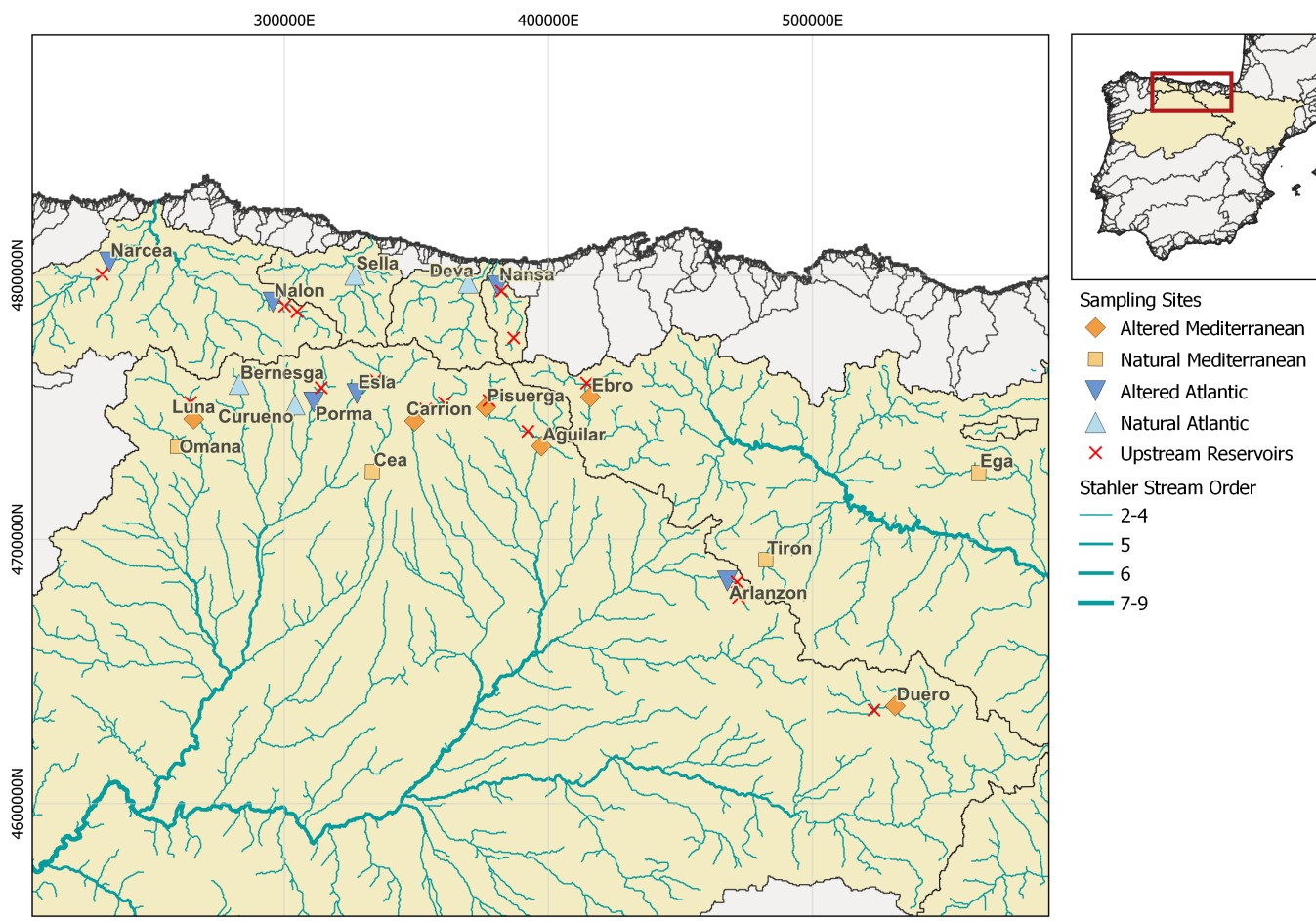

**Figure 2.** Location of sampling sites in selected rivers with Atlantic (blue) and Mediterranean (orange) (reference) flow regimes in Northern Spain. Light and dark shades of the colour represent rivers with natural and altered flow regimes, respectively. All sampling locations had spatially independent, non-nested catchments except Aguilar nestled in Pisuerga. The rivers and their basins were sourced from the HydroSHEDS database (Lehner and Grill, 2013)

The selected altered rivers all have a dam with a reservoir (predominantly in the headwaters or upper reaches), although the exact alteration of the flow regime may vary depending on the size, purpose and operation of the involved dam (Table 1). The sampling sites in the altered rivers were located on average 4.1 km (range 1.74 - 33.6 km) downstream of a reservoir and some sites had multiple upstream reservoirs. The catchment areas upstream of the sampling sites ranged from 89 km$^2$ to 1258 km$^2$ (Table 1) with an average of 429 km$^2$. All the reservoirs in altered Mediterranean rivers were used for irrigation, while in altered Atlantic rivers three reservoirs had irrigation and three reservoirs had hydropower production as their main purpose. Only one hydropower-producing reservoir, Narcea, has hydropeaking at short timescale and insufficient capacity for seasonal

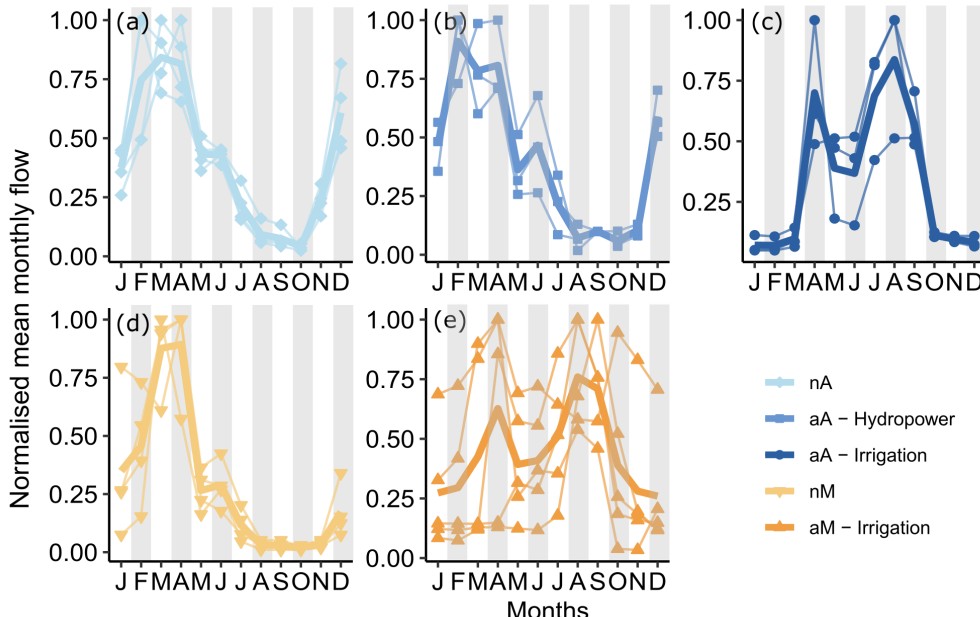

**Figure 3.** Normalized (to maximum) mean monthly flow of each river grouped by flow regime (a) natural Atlantic (nA) in light blue, (b, c) altered Atlantic (aA) in dark blue shades according to their reservoir purpose, (d) natural Mediterranean (nM) in light orange, and (e) altered Mediterranean (aM) in dark orange (only reservoirs with irrigation purpose). The thick line represents the average of the flow regimes. The shaded months are the sampling months.

storage. On the other end of the spectrum, while Esla has the biggest reservoir capacity, Ebro has the highest storage index (i.e., highest capacity compared to its average annual runoff). In each river, we characterized DOM properties bi-monthly from October 2017 to August 2018 totalling 6 sampling occasions (October and December 2017, and February, April, June and August 2018). Discharge measurements for each river and sampling day are given in Supp. Table S1.

### 2.2 DOM composition

We measured DOC concentration, DOM absorbance and fluorescence, and DOM molecular size distribution. In the field, we filtered water samples through pre-combusted (450 $^o$C, 4 h) glass fibre filters of 0.3 $\mu$m nominal pore size (Advantec GF-75, Japan) into acid-washed, MilliQ water-rinsed, pre-combusted (450 $^o$C, 4 h) glass vials. We stored all the samples in the dark at 4 $^o$C until analysis. We measured DOC concentration (mg L$^{-1}$) by high-temperature catalytic combustion of an acidified and sparged water sample with a Shimadzu TOC-V Analyser with a typical limit of quantification (computed as 6 times the standard deviation of blanks) of < 0.1 $\pm$ 0.08 mg L$^{-1}$ and a typical analytical precision of 1.5 %. To prepare standards for DOC-measurements we used potassium hydrogen phtalate (KHP). We took DOM absorbance and fluorescence measurements simultaneously using an Aqualog (Horiba Ltd, Kyoto, Japan). We used MQ-water as a blank inserted every 10-15 measurements; each of 3 field replicates was measured 3 times and spectra were visually checked for consistency and

outlier removal. We measured absorbance in 5 nm increments from 250 nm to 600 nm and fluorescence as excitation-emission-matrices (EEMs); we excited the samples from 250 nm to 600 nm in 5 nm increments with emission from 212 nm to 620 nm in 2 nm increments. We subtracted MilliQ water EEMs from all sample EEMs to remove optical scatter phenomena but did not apply any inner filter correction since the decadal absorption coefficient at 254 nm ($A_{254}$) was lower than the threshold of 0.3 for all samples (Ohno, 2002).

We used the Aqualog measurements to calculate 6 optical indices; fluorescence index (FI), humification index (HI), freshness index ($\beta/\alpha$), E2:E3 ratio, slope ratio (SR) and specific UV absorbance ($SUVA_{254}$), as summarised in Table 2. Further, we performed a parallel factor analysis (PARAFAC) after Raman-normalizing, smoothing and normalizing each EEM by setting the maximum emission to 1 (Coble et al., 2019). We used the StaRdom package (Pucher et al., 2019) in R version 4.0.3 (R Core Team, 2020) to partition the EEMs into individual components following the PARAFAC modelling strategy. We chose the 8-component PARAFAC model as the one achieving the best representation of the 699 EEMs dataset. We verified the model with split-half analysis with random sample selection following residual analysis. Sample-specific fluorescence of components was expressed in Raman units by reverting the normalization step through the multiplication of $F_{max}$ values with the original EEMs´ maximum emission values. We scanned the literature via the OpenFluor database (Murphy et al., 2013) for similar components (Table3). The components C1, C2, C3, C4 and C5 were reported as old humic-like DOM with strong terrestrial signals composed of mostly high MW substances. C6, C7 and C8 were all protein-like compounds of recently produced fresh material (Table 3). We grouped the relative contribution of the first 5 components (C1-C5) into an indicator of humic and terrestrial signatures ($C_{Terrestrial}$) by dividing their sum of relative contributions by the sum of all components. In the same manner, we turned the last 3 components into an indicator of freshly produced microbial input ($C_{Microbial}$).

**Table 2.** Calculated optical indices, their meaning and used method

| Indice | Description | Method |
|--------|-------------|--------|
| FI | Fluorescence index; the relative contribution of terrestrial to microbial fluorophores | Ratio of emissions of 450 and 500 nm at 370 nm excitation (McKnight et al., 2001) |
| HI | Humification index | Dividing the sum of emission intensities from 435 to 480 nm by the sum of intensities from 300 to 345 nm and from 435 to 480 nm (Ohno, 2002) |
| $\beta/\alpha$ | Freshness index, a proxy for relative microbial (fresh) contribution | Ratio of emissions at 380 nm to the maximum emission between 420 and 435 nm at 310 nm excitation (Harjung et al., 2019; Wilson and Xenopoulos, 2009) |
| E2:E3 | Inverse indicator of molecular size | Absorbance ratio of 250 to 365 nm |
| SR | Slope ratio; inverse indicator of molecular weight | Slope ratio of short slope between 275 and 295 nm and long slope between 350 nm and 400 nm (Loiselle et al., 2009) |
| $SUVA_{254}$ | Aromaticity indicator | Decadal UV absorbance at 254 nm divided by DOC concentration |

We used liquid size–exclusion chromatography (SEC) with organic carbon and nitrogen detection (LC-OCD-OND; Huber et al. (2011)) for the molecular size distribution measurements. This allowed us to estimate the abundance of non-humic high

**Table 3.** PARAFAC components with the modelled maximum emission and excitation values ($Ex_{max}$, $Em_{max}$) were compared to the literature using OpenFluor (TCC combined > 0.90). For each component, the main sources, the peak names and selected keywords from the literature are given.

| Source | Comp. | $Ex_{max}$ | $Em_{max}$ | Keywords |
|---|---|---|---|---|
| | C1 | 330 | 444 | Terrestrial humic-like, Peak C (Kothawala et al., 2012), high aromaticity, high MW, photosensitive (Lambert et al., 2016a) |
| | C2 | <260 (340) | 520 | Terrestrial, humic-like DOM, high MW, aromatic (Kothawala et al., 2012; (Lambert et al., 2016a, b), Peak A and Peak C (Lin and Guo, 2020) |
| $C_{Terrestrial}$ | C3 | <325 | 390 | Humic-like, low MW, UV-A, fresher DOM, Peak M (Kothawala et al., 2012) |
| | C4 | 260 (300) | 420 | High MW, aromatic (Kothawala et al., 2012), photoproduct of a terrestrial, Peak A, Terrestrial humic-like (Lambert et al., 2016b) |
| | C5 | <260 (380) | 460 | High MW (Kothawala et al., 2012), photochemically degradable, terrestrial humic-like (Harjung et al., 2019), aromatic, fulvic acid (Yang et al., 2017) |
| | C6 | <260 (295) | 380 | Protein-like, microbial-humic like fluorescence (Lambert et al., 2016a), tryptophan-like (Derrien et al., 2018), microbially produced, Peak T/Peak M mixture (Kothawala et al., 2012) |
| $C_{Microbial}$ | C7 | 260 | 340 | Tryptophan-like recent biological production (Lambert et al., 2016a), Protein-like (Lambert et al., 2017) |
| | C8 | 270 | 310 | Protein-like recent biological activity fluorophores (Lambert et al., 2016a), tyrosine-like fluorophores (Painter et al., 2018) |

molecular weight substances ($C_{HMWS}$, like polysaccharides and proteins, mg C L$^{-1}$); humic substances ($C_{Humic}$, mg C L$^{-1}$) and neutral, hydrophilic to amphiphilic low molecular weight substances ($C_{LMWS}$, e.g., aldehydes, sugars and amino acids, mg C L$^{-1}$). For samples below the detection limit, we used half the value of the detection limit in the final dataset to allow
145 multivariate data analysis. To have an estimate of the relative contributions of each of the defined size fractions, we divided each by total dissolved organic carbon concentration measured by SEC (Heinz and Zak, 2018).

## 2.3 Statistical analysis

To assess differences among the flow regimes, we used principal component analysis (PCA) using the 81 flow indices reported in Peñas et al. (2016) and grouped the rivers according to their flow regime to see if flow alterations homogenize or diversify
150 natural flow characteristics (Supp Fig. S1). To this aim, we also used the average distance to the centroid of a particular flow regime as a measure of dispersion in the same way as for DOM data (see below for assessment of the among-river variation of annual mean DOM composition). The magnitude of 1-30-90 day High Flow Events (1HF, 30HF, 90HF) and the magnitude of 1-30-90 day Low Flow Events (1LF, 30LF, 90LF) were selected to compare the four flow regimes following the findings of natural and altered rivers' flow differences of Goldenberg-Vilar et al. (2022) on the same rivers.

As measures of mean DOM concentration and mean composition for each river, we computed annual means of the DOC concentration, of optical indices and of SEC fractions. To express the seasonal variability of these variables we computed river-specific temporal coefficients of variation (CVs) and used them as indicators of annual shifts in DOM composition. Annual means and temporal CVs were then used as univariate responses describing DOM regimes to test for differences between the four flow regimes. For this, we followed a 1-way approach in two steps for each response variable: First, we tested mean and CV differences of all four flow regimes with a one-way test (Welch, 1951), which - unlike classical ANOVA - does not need variance homogeneity as a prerequisite. In case of significance, this was then followed by planned paired t-tests between (i) natural Atlantic and natural Mediterranean (nA vs nM), (ii) altered Mediterranean and natural Mediterranean (aM vs nM), (iii) altered Atlantic and natural Atlantic (aA vs nA). We corrected p-values with the Bonferroni method assuming three a-priori planned tests. We also compared the spatial variation of annual means and temporal CVs across the rivers of the four flow regimes. For this, we followed a similar 1-way approach in two steps, where we first tested for variance homogeneity of annual means and of temporal CVs across all four regimes with a Bartlett test, which - when significant - was followed by planned F-tests between (i) nA vs nM, (ii) aM vs nM and (iii) aA vs nA. Taken together, our response variables included measures of mean location and temporal variation for each river´s DOM indicators, which serve as DOM regime descriptors, and measures of (spatial) variation of these DOM regime descriptors across rivers belonging to the same flow regime. The latter was specifically computed to assess whether, in any given natural reference flow regime, dam-induced flow alteration diversifies DOM regimes across rivers compared to DOM regimes under natural flow.

Additionally, to assess the similarities and differences in annual average DOM composition and shifts in DOM composition between natural and altered rivers in a more integrative, multivariate way, we did a principal component analysis (PCA) using the relative fluorescence of PARAFAC components, optical indices and SEC fractions. We looked for recognizable differences in DOM composition among flow regimes (i.e., between natural and altered regimes of both hydrological classes) with regard to location (mean DOM composition) and specifically with regard to temporal shifts in DOM composition in the various rivers. To statistically test for differences in DOM composition among flow regimes we used a 1-way permutational multivariate ANOVA (Anderson, 2001) on annual mean locations (river-specific centroids) in the PCA-space with the same paired testing strategy as described above. Temporal shifts in DOM composition are graphically represented in 2D PCA-space by polygons for each river (Fig. 6c-f), where each corner represents the DOM composition of a sampling date in the PCA space and the centroid represents the mean DOM composition of that river over the entire sampling period. The average distance of individual sampling dates to the centroid of a river serves as a measure of temporal shifts in DOM composition; it is computed as a dispersion (i.e., multivariate variation) on all dimensions of the PCA using the function betadisper() from the vegan R-package (Oksanen et al., 2025). Here, high average distance to the centroid (high dispersion) means strong temporal shifts in DOM composition. To statistically test the temporal shifts in DOM composition we followed the same analysis approach as described above for a univariate response variable, i.e. a one-way ANOVA comparing dispersion as a response among all four flow regimes followed by three planned t-tests as post hoc tests.

In analogy to the analysis of univariate responses, we also compared the among-river variation of annual mean DOM composition and that of temporal shifts in DOM composition across the four flow regimes given by the multivariate dataset. Our

motivation was again to assess if, in any given natural reference flow regime, dam-induced flow alteration diversifies DOM regimes (in either mean location or in temporal shifts of composition, i.e. dispersion) compared to DOM regimes under natural flow. Again, we followed a similar 1-way approach in two steps. For the among-river variation of annual mean DOM composition, we tested for dispersion homogeneity of river centroids across all four regimes with PERMDISP, a permutational equivalent of the Bartlett test (Anderson, 2006), which - when significant - was followed by planned permutational F-tests between (i) nA vs nM, (ii) aM vs nM and (iii) aA vs nA. For the among-river variation of temporal shifts in DOM composition, we used the river-specific measure of temporal dispersion as a univariate response and followed the same approach as described above for variation of univariate responses (Bartlett-test followed by planned t-tests).

As the individual axes of the PCA can be understood as distinct meaningful components of a river´s DOM regime, we also tested the differences in average location and variance along PC1 and PC2 axes following the same univariate strategy of a one-way test followed by planned t-tests as described above. Similarly, we compared across-river variance of average location and variance along the PC1 and PC2 axis, with a Bartlett test followed by planned F-tests as post hoc analysis. A summary of the statistical workflow followed is included in Fig. 1.

To look for links between characteristics of the flow regime and temporal shifts in DOM composition, we performed three partial least squares (PLSR) analyses (one for the overall shifts in DOM composition, one each for PC1 and PC2 variance). We chose PLSR analysis due to its reputed good performance in situations with high collinearity of predictors and small sample size (Carrascal et al., 2009; Coble et al., 2019). As predictors, we used 81 hydrological indices computed by Peñas and Barquín (2019) that were grouped into five flow regime components: magnitude, duration, frequency, timing, and rate of change (Supp. Table S3). As dependent variables we used the multivariate dispersion (for the overall shifts in DOM composition) and the two variances on the first two PCA axes for each river, which can be understood as the two most important components of shifts in DOM composition, each having a specific qualitative character captured by the respective PCA-axis. In PLSR analysis, we selected the hydrological indices with VIP-scores (Variable Importance on the Predictor) higher than 1 as the most important explanatory variables and reported only the first component of the PLSR analysis for simplicity.

## 3 Results

### 3.1 Flow Regimes

Both Atlantic and Mediterranean natural flow regimes are characterized by high flows in early to late spring and low flows throughout the summer with a minimum in August-September before fall rains begin (Fig. 3a and d). Additionally, natural Atlantic flow regimes have a second high flow peak in early winter (November-December) compared to the natural Mediterranean regime.

Contrarily, altered Mediterranean rivers, with their irrigation reservoirs, do not experience the natural succession of high spring flows followed by summer droughts as observed in the natural Mediterranean rivers (Fig.3b and c). From previous studies, we know that our altered Mediterranean rivers collect water during winter and spring and release it from May to September (Pompeu et al., 2022), which is the reason behind the high summer flows. We also know that this behaviour is

limited by the storage capacity of these reservoirs, where a lower storage capacity requires an additional 'pre-release' in late winter-spring due to the capacity being reached, resulting in a double-humped shape of the annual hydrograph (Fig. 3c and e).

When the mean of the altered Atlantic rivers is compared to its natural equivalent (Fig. 3a-c), we see that the above-mentioned introduction of summer high flows is not as consistent, i.e. three rivers have distinct summer high flows while three others have late summer droughts. When grouped by reservoir purpose, rivers with hydropower reservoirs (Fig. 3b, Narcea, Nalon, Nansa) still possess summer droughts even though they have altered flow regimes, i.e., they somewhat mimic the natural flow regime expected of that river. Contrarily, rivers with irrigation reservoirs (Fig. 3c, Porma, Esla, Arlanzon) have clear

summer high flows and a secondary peak in spring due to limited storage capacity. Thus, in altered Atlantic rivers, flow regime heterogeneity among the rivers comes from the different purposes of the reservoirs (irrigation vs hydropower), whereas in altered Mediterranean rivers, flow regime heterogeneity is derived mostly from the differences in storage capacity of reservoirs exclusively dedicated to irrigation.

From previous studies on our rivers, we know that altered rivers show a higher diversity of flow regimes compared to natural

rivers (Goldenberg-Vilar et al., 2022). In a PCA analysis, we found that altered Mediterranean rivers have 13-fold and altered Atlantic rivers 44-fold higher dispersion in the PCA space compared to their natural equivalents, while both natural flow regimes showed similarly small dispersions (Supp Fig. S1). This confirms that altered Atlantic regimes encompassing both irrigation and hydropower reservoirs are more diversified than altered Mediterranean regimes with irrigation reservoirs only.

As a final step, we looked at selected flow regime characteristics (Fig. 4), specifically at the magnitude of 1-30-90 Day High

and Low Flow Events (selected from Goldenberg-Vilar et al. (2022)). These selected indicators are used to show the general trend of our flow regimes in magnitude and duration of annual minimum and maximum flows, covering short-duration flood pulses as well as seasonal droughts. Both natural flow regimes have higher 1-Day High Flows than their altered counterparts. Natural Mediterranean rivers have higher 30-Day High Flows than altered Mediterranean rivers, whereas this difference was not as clear between natural and altered Atlantic rivers. There was not much visible difference between regimes in 90-Day

High Flows. Similarly, natural Atlantic rivers have lower 1-Day Low Flows than Atlantic rivers in flows altered by irrigation dams but not in those with flow altered by hydropower dams. Natural rivers have lower 30-Day Low Flows and 90-Day Low Flows than altered counterparts for both Atlantic and Mediterranean rivers.

## 3.2    Average and temporal variation of DOC concentration

The DOC concentration varied 10-fold from 0.59 mg $L^{-1}$ to 5.3 mg $L^{-1}$ across rivers throughout the year. Planned t-tests did

not show any significant difference in annual mean DOC concentration (Fig. 5a, Table 4) between the two natural flow regimes (nA-nM), nor between the altered and natural rivers (aA-nA and aM-nM, respectively). The consistently higher median in all altered Atlantic rivers compared to the natural Atlantic rivers and partly significant results prior to Bonferroni correction point to a lack of statistical power due to low sample size causing a lack of significance. The two highest average DOC concentrations belong to Duero and Ebro, respectively (Figure 5a). We could not find any common hydrological characteristics

separating these two from the remaining altered Mediterranean rivers, but note that Ebro has the highest storage index in the dataset. We also could not find any association of DOC concentration with discharge (Supp Table 1). In fact, Ebro and Duero

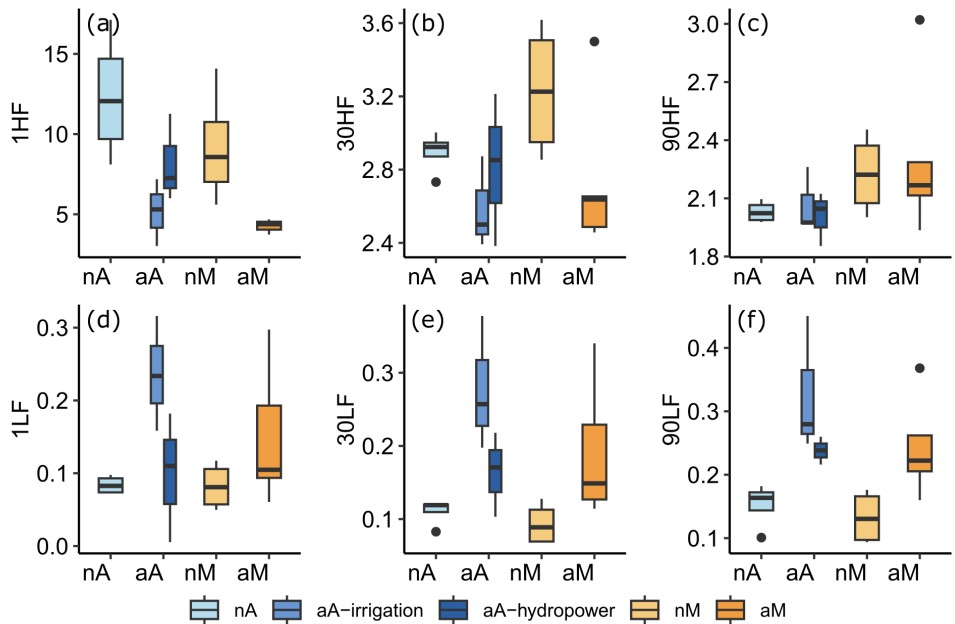

**Figure 4.** Boxplots of magnitude of extreme flow events representing flow variability; Magnitude of (a) 1-Day High Flow Event (1HF), (b) 30-Day High Flow Event (30HF), (c) 90-Day High Flow Event (90HF) and (d) 1-Day Low Flow Event (1LF), (e) 30-Day Low Flow Event (30LF), (f) 90-Day Low Flow Event (90LF). Natural Atlantic (nA) rivers are shown in light blue, altered Atlantic (aA) in dark blue, separated by different shades according to their dam purpose; natural Mediterranean (nM) rivers are shown in light orange and altered Mediterranean (aM) in dark orange, where all rivers are altered by dams built for irrigation.

show completely opposite trends; Ebro has its highest DOC concentration in April at 1.05 $m^3$/s and lowest DOC value in August at 5.31 $m^3$/s, which coincides with the water release timings. Contrarily, Duero has its highest DOC concentration in June at 4.56 $m^3$/s and lowest DOC concentration in December at 0.88 $m^3$/s, when reservoirs are filled up.

The annual DOC variation of the rivers ($CV_{DOC}$) ranged from 14% to 53%. The two natural regimes (nA-nM) showed no significant difference in $CV_{DOC}$ (Table 4). Looking at the change in DOC through the sampled months in Atlantic rivers (Fig. 5b), we observe that both altered and natural rivers' average DOC time-series follow similar paths through the year with lowest values in August followed by an increase until spring high flows. In Mediterranean rivers, however (Fig. 5c), altered rivers' DOC time-series increase in April and June, while most natural rivers have low values in these months.

In both flow regimes, the among-river variation of DOC and $CV_{DOC}$ seems lower in natural rivers than in their altered equivalents, but we could not find any significant difference between any of the tested flow regime pairs (data not shown), likely again because of low statistical power.

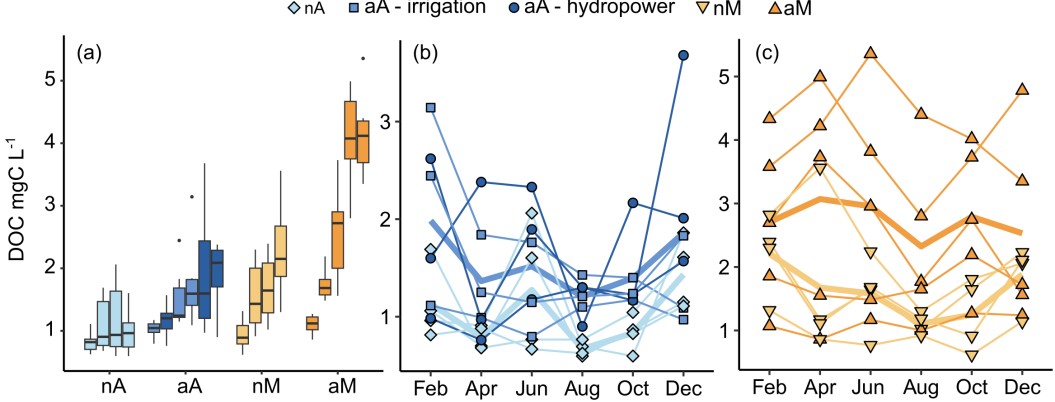

**Figure 5.** (a) DOC of each river as individual boxplots grouped by flow regime; natural Atlantic (nA) in light blue, altered Atlantic (aA) in dark blue, natural Mediterranean (nM) in light orange and altered Mediterranean (aM) in dark orange. (b) Atlantic and (c) Mediterranean rivers' temporal DOC behaviours are represented as lines following the same colour scheme. The aA rivers are separated into two reservoir purpose groups, irrigation and hydropower. All aM rivers have irrigation reservoirs. The individual points represent single DOC measurements in various rivers of each flow regime in each sampling month, thus thin lines represent the trajectory of a river throughout the year. The thick lines connect the monthly averages for all the rivers of the respective flow regimes.

### 3.3    Average DOM composition and its temporal shifts

In all regimes, DOM showed strong terrestrial characteristics throughout the year with relatively low FI (<1.4) and $\beta/\alpha$ (<1)
and high HI ($0.86 \pm 0.04$) and SUVA$_{254}$ (>2.4). Also, with values >60%, the indicator C$_{Terrestrial}$ showed dominance of soil-derived materials over microbially produced materials (McKnight et al., 2001; Fellman et al., 2010; Hansen et al., 2016). The FI values ranged from 1.09 to 1.38 among the rivers throughout the year, typical for natural waters (Hansen et al., 2016). The SR values were relatively low for all flow regimes (SR<1) indicating dominance of higher MW compounds and aromatic, plant-derived materials (Oliver et al., 2016). The relative contribution of SEC fractions also indicates that, in all of the regimes,
humic substances were most abundant (C$_{HS}$), followed by simple monomers like sugars and amino acids (C$_{LMWS}$) and polysaccharides and proteins (C$_{HMWS}$).

When we compare the two natural regimes of Atlantic and Mediterranean rivers, there was no significant difference in any annual average optical indicator or SEC fraction (Table 4). However, CV$_{SR}$ and CV$_{E2:E3}$ were significantly higher in natural Atlantic rivers compared to natural Mediterranean rivers. The F-test results for all annual averages and compositional shift
indicators show similar among-river variances between the two natural regimes (nA-nM, data not shown).

When we compare the natural regimes to their altered equivalents in terms of average DOM composition, we could not observe any significant difference in the annual average optical indices (Table 4) in Atlantic or Mediterranean rivers. Among the CV values, indicating shifts in DOM composition, we observed that in Atlantic rivers, CV$_{E2:E3}$, CV$_{CProtein}$ and CV$_{CTerrestrial}$ were significantly higher in natural rivers compared to their altered equivalents (Table 4). In Mediterranean rivers, CV$_{SR}$ was

significantly higher in altered rivers compared to their natural equivalents. From the F-test results, we observe that all of the average DOM composition and compositional shift indicators showed similar among-river variances between the natural and altered regimes (data not shown).

We further analysed the similarities and differences in the annual average and temporal shifts in DOM composition in a multivariate approach with a PCA. The first and second axes of the PCA (Fig. 6, PC1) explained 47 % and 16 % of the total variation of DOM composition. PC1 was negatively correlated with terrestrial and humic components ($C_{Terrestrial}$), humic substances ($C_{HS}$) and HI, and positively correlated with microbially sourced components ($C_{Microbial}$), FI, $\beta/\alpha$, and $C_{HMWS}$, representing non-humic biopolymers (Huber et al., 2011). PC2 was negatively correlated with SR, E2:E3 and SUVA$_{254}$ and positively correlated with $C_{LMWS}$. Taken together, the PCA space from the bottom left to the top right corner shows a gradient of terrestrially sourced, aromatic, high MW and presumably diagenetically older material to microbially sourced, fresher, low MW material. Notably, PC1 and PC2 capture indicators of DOM composition described with different methods: PC1 groups mostly fluorescence-based indices ($C_{Microbial}$, $C_{Terrestrial}$, FI, HI, $\beta/\alpha$) and PC2 groups absorbance indices (E2:E3, SUVA$_{254}$, SR) and SEC fractions. The patterns in PCA space were not related to DOC concentration or discharge (data not shown).

We used the centroid location and average distance to the centroid of each river (represented as polygons in 2D space in Fig. 6c-f) as proxies for the annual average and temporal dispersion of DOM composition, respectively. We could not find a significant difference in centroid location, i.e. annual average DOM composition, between any of the compared pairs of flow regimes (Supp. Table S2). In terms of temporal dispersions, we could not identify a significant difference between the natural rivers nor between the natural and altered Mediterranean rivers, yet we observed that natural Atlantic rivers have a significantly higher average distance to the centroid compared to their altered Atlantic equivalents (Fig. 6c-d, Fig. 7a, Supp. Table S2). Thus, in Atlantic rivers, DOM composition was more variable over time when the flow regime was natural. This effect was not as clear in Mediterranean rivers: Though DOM regimes appeared as being temporally less variable in all altered Mediterranean rivers, they showed a higher diversity in natural rivers. This higher diversity of DOM regimes in natural Mediterranean rivers, with some being naturally quite dynamic while others show fairly stable DOM composition in time, precluded the identification of a significant flow alteration effect. Notably, however, Mediterranean rivers showed slightly more clearcut flow alteration effects along the individual PCA-axes (see below), with slightly more prominent excursions towards higher PC1 scores and lower PC2 scores in dam-impacted rivers towards late summer and fall.

The global tests comparing the four flow regimes did not show any significant difference in among-river variance regarding the mean centroid location and the mean dispersions from the river centroids (Supp. Table S2). Still, DOM regimes did not appear similarly variable among the various flow regimes: Specifically, the seasonal shifts in DOM composition in natural Mediterranean rivers were quite variable. Here, by optical assessment, dams caused a homogenization across rivers with invariable DOM through seasons (Fig. 6e-f). Also, seasonal shifts in DOM composition in natural Mediterranean regimes seemed to be more variable than in Atlantic counterparts.

Since individual PCA axes represent qualitatively distinct shifts in DOM composition, we looked at the timelines, average locations and variances of rivers along the PC1 and PC2 axes separately as well. Timelines of DOM composition throughout

**Table 4.** Means of river-specific annual means and temporal coefficients of variation (CV) for each of the four flow regimes; natural Atlantic (nA), altered Atlantic (aA), natural Mediterranean (nM), altered Mediterranean (aM). The table provides means ± standard deviations. Superscripted letters above the values of mean and standard deviation indicate significant differences according to t-tests and F-tests done pairwisely between two flow regimes. These letters allow pairwise comparisons between any two flow regimes, but Bonferroni-correction was applied to p-values assuming only 3 a-priori planned tests of relevance (nA vs. nM, nA vs. aA, nM vs. aM). The letters were dropped when the pairwise comparisons did not indicate a significant difference. For the calculations of the mean and CV values, 6 measurements distributed across one year were taken into account. The number of sites (n) for each flow regime is given in the column headlines.

| | Atlantic | | Mediterranean | |
| --- | --- | --- | --- | --- |
| | natural (n = 4) | altered (n = 5) | natural (n = 4) | altered (n = 6) |
| *Water Chemistry* | | | | |
| DOC (mg C L$^{-1}$) | $1.02^A \pm 0.15$ | $1.55^{AB} \pm 0.39$ | $1.62^{AB} \pm 0.56$ | $2.73^B \pm 1.37$ |
| CV$_{DOC}$ | $0.38 \pm 0.13$ | $0.33 \pm 0.14$ | $0.33 \pm 0.04$ | $0.19 \pm 0.07$ |
| *Optical Indices* | | | | |
| FI | $1.20 \pm 0.02$ | $1.18 \pm 0.04$ | $1.23 \pm 0.04$ | $1.22 \pm 0.03$ |
| CV$_{FI}$ | $0.04 \pm 0.01$ | $0.03 \pm 0.005$ | $0.03 \pm 0.02$ | $0.03 \pm 0.01$ |
| HI | $0.86 \pm 0.02$ | $0.86 \pm 0.02$ | $0.88 \pm 0.02$ | $0.84 \pm 0.03$ |
| CV$_{HI}$ | $0.06 \pm 0.03$ | $0.03 \pm 0.009$ | $0.03 \pm 0.01$ | $0.05 \pm 0.02$ |
| $\beta/\alpha$ | $0.66 \pm 0.03$ | $0.66 \pm 0.05$ | $0.66 \pm 0.02$ | $0.70 \pm 0.02$ |
| CV$_{\beta/\alpha}$ | $0.07 \pm 0.01$ | $0.07 \pm 0.03$ | $0.05 \pm 0.02$ | $0.09 \pm 0.03$ |
| E2:E3 | $5.80^A \pm 0.05$ | $6.44^{AB} \pm 0.06$ | $6.30^A B \pm 0.04$ | $7.10^B \pm 0.06$ |
| CV$_{E2:E3}$ | $0.31^A \pm 0.10$ | $0.13^B \pm 0.04$ | $0.13^{\ B} \pm 0.03$ | $0.14^{\ B} \pm 0.04$ |
| SR | $0.89 \pm 0.07$ | $0.89 \pm 0.10$ | $0.82 \pm 0.05$ | $0.97 \pm 0.09$ |
| CV$_{SR}$ | $0.34^A \pm 0.05$ | $0.20^B \pm 0.10$ | $0.11^{\ B} \pm 0.02$ | $0.20^{AB} \pm 0.05$ |
| SUVA$_{254}$ | $2.93 \pm 0.32$ | $2.74 \pm 0.37$ | $2.63 \pm 0.26$ | $2.40 \pm 0.33$ |
| CV$_{SUVA254}$ | $0.18 \pm 0.04$ | $0.20 \pm 0.05$ | $0.23 \pm 0.06$ | $0.27 \pm 0.05$ |
| *PARAFAC Components* | | | | |
| C$_{Terrestrial}$ | $0.69 \pm 0.02$ | $0.69 \pm 0.04$ | $0.71 \pm 0.05$ | $0.64 \pm 0.07$ |
| CV$_{CTerrestrial}$ | $0.19^A \pm 0.07$ | $0.09^{\ B} \pm 0.02$ | $0.12^{AB} \pm 0.04$ | $0.12^{AB} \pm 0.04$ |
| C$_{protein}$ | $0.31 \pm 0.02$ | $0.31 \pm 0.04$ | $0.29 \pm 0.05$ | $0.36 \pm 0.07$ |
| CV$_{Cprotein}$ | $0.40^A \pm 0.10$ | $0.20^B \pm 0.05$ | $0.28^{AB} \pm 0.04$ | $0.21^B \pm 0.05$ |
| *Relative Contribution of SEC fractions* | | | | |
| C$_{HMWS}$ | $0.09 \pm 0.02$ | $0.06 \pm 0.01$ | $0.07 \pm 0.02$ | $0.09 \pm 0.03$ |
| CV$_{CHMWS}$ | $0.60 \pm 0.25$ | $0.45 \pm 0.27$ | $0.59 \pm 0.22$ | $0.65 \pm 0.18$ |
| C$_{LMWS}$ | $0.13 \pm 0.01$ | $0.13 \pm 0.02$ | $0.14 \pm 0.02$ | $0.14 \pm 0.01$ |
| CV$_{CLMWS}$ | $0.27 \pm 0.08$ | $0.25 \pm 0.10$ | $0.27 \pm 0.07$ | $0.11 \pm 0.10$ |
| C$_{HS}$ | $0.77 \pm 0.03$ | $0.81 \pm 0.07$ | $0.79 \pm 0.03$ | $0.76 \pm 0.04$ |
| CV$_{CHS}$ | $0.12 \pm 0.03$ | $0.09 \pm 0.09$ | $0.11 \pm 0.06$ | $0.07 \pm 0.02$ |

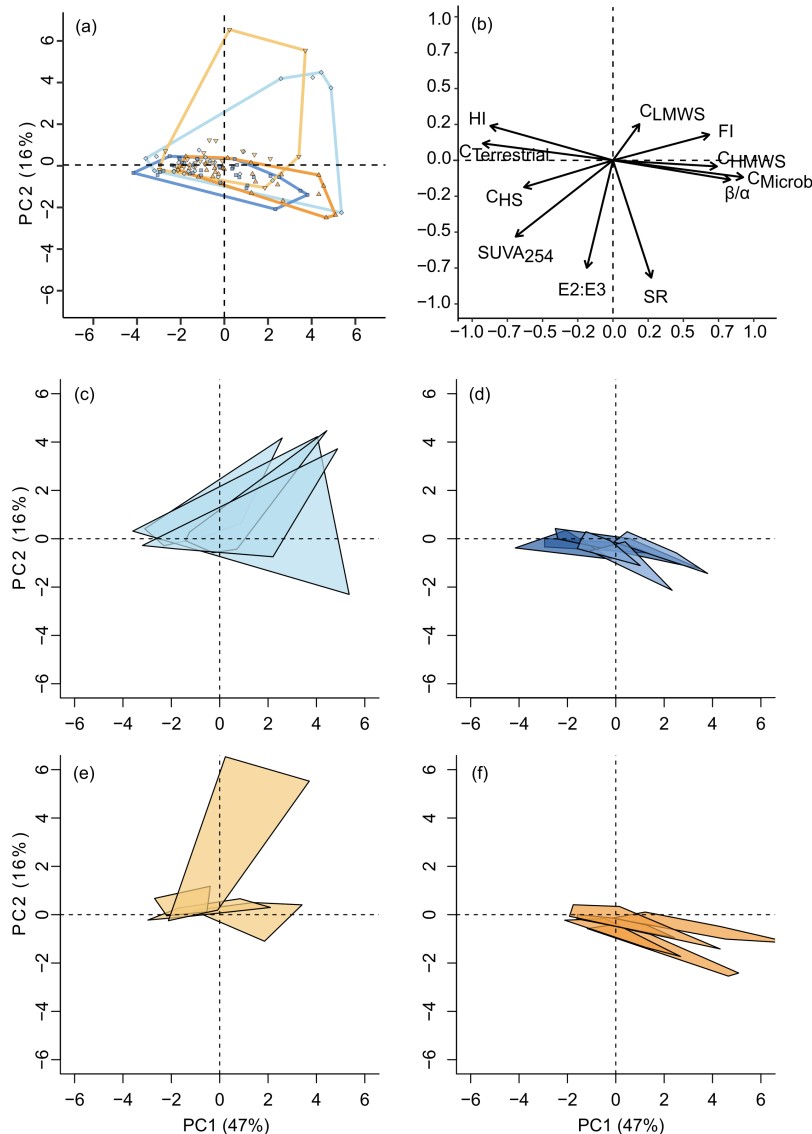

**Figure 6.** Principal component analysis of DOM composition. Graphical representation of (a) scores and (b) loadings of the PCA results where scores of each sampling occasion show a large overlap of flow regimes regarding DOM composition. The bottom panels show the seasonal shifts in DOM composition in each river as a polygon in the same PCA space, where (c) the natural Atlantic rivers (nA) are represented in light blue, (d) the altered Atlantic (aA) rivers are represented in dark blue, (e) the natural Mediterranean (nM) rivers are represented in light orange and (f) the altered Mediterranean (aM) rivers are represented in dark orange. The PCA is based on absorbance and fluorescence measurements; humic-like components ($C_{Terrestrial}$), protein-like components ($C_{Microbial}$), specific absorption at 254 nm ($SUVA_{254}$), humification index (HIX), fluorescence index (FI), freshness index ($\beta/\alpha$), slope ratio (SR), inverse molecular weight indicator (E2:E3) and size exclusion chromatography fractions: high MW biopolymers ($C_{HMWS}$), humic substances ($C_{HS}$) and low MW substances ($C_{LMWS}$).

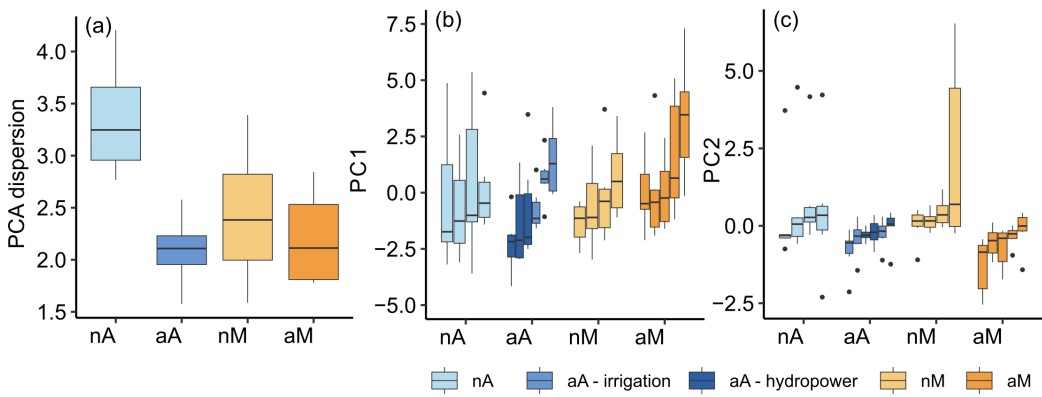

**Figure 7.** Seasonal shifts in DOM composition as average dispersion (distance to river centroid) along all the PCA axes (a); temporal variation of DOM in each river along PC1 (b) and PC2 (c). The rivers are grouped by flow regime; natural Atlantic (nA, light blue), altered Atlantic (aA, dark blues), natural Mediterranean (nM, light orange) and altered Mediterranean (aM. dark orange) rivers. The aA rivers are separated into two reservoir purpose groups, irrigation and hydropower. All aM rivers have reservoirs used for irrigation.

the year showed similar patterns on PC1 for all flow regimes (Supp Fig. 2a-b). Contrarily, on PC2, natural rivers differ from their altered equivalents both in August and October (Supp Fig. 2c-d). Plotting the $\beta/\alpha$ index as a particular example, it turned out higher in October in altered Mediterranean rivers than in its natural equivalents; such a difference was not observed in other months nor in Atlantic rivers (Supp Fig. 2 e-f). We could not find any significant difference in average location along the PC1 or PC2, neither between the two natural regimes nor between altered and natural Mediterranean rivers (Fig. 7b, Supp Table 1).

However, we can visually see that Atlantic rivers with irrigation dams have the three highest medians on PC1, while Atlantic rivers with hydropower dams have the three lowest.

We found that altered Atlantic rivers (Supp. Table S2) have lower average PC2 scores than natural Atlantic rivers. We could not find any significant difference in temporal variation along the PC1 and PC2, neither between the natural regimes nor between the altered and natural Mediterranean rivers (nM-aM, Fig. 7b, Supp. Table S2). However, we found that natural

Atlantic flow regimes have significantly higher seasonal shifts in DOM composition than their altered equivalents along both PC1 and PC2 axes (nA-aA, Supp. Table S2).

Moreover, along PC1, the among-river variation of mean DOM was similar among the rivers. Along PC2 we found a difference between the natural rivers (nA-nM; Supp. Table S2). Similarly, along PC1, among-river variation of seasonal shifts in DOM composition was similar for all the rivers. Contrarily, along PC2, it was different both between the natural rivers and

between natural and altered Atlantic Rivers (nA-nM, nA-aA; Supp. Table S2). These results confirm our visual assessments (Fig. 6c-f, Fig. 7).

## 3.4 Linking DOM regimes to flow regimes

The first component of the partial least square regression (PLSR) models explained 30.9 % of the variance of the dispersion in the entire multidimensional PCA space, 18.7% of the variance on the PC1 axis, and 24.0 % of the variance on the PC2 axis. For ease of interpretation, only the $1^{st}$ component was selected from each of the three PLSR models. Based on VIP > 1 as a criterion, 34, 34, and 29 indices were selected to explain the multidimensional dispersion and the variance of PC1 and PC2 axes, respectively. Mostly, indices identified as important for "Multidimensional dispersion" - as the global model of shifts in DOM composition - were then also found as relevant for explaining variances of PC1 or PC2 or both axes. A few indices, however, were identified as relevant in only one of the three models, suggesting multivariate dispersion to be a maybe noisy but more comprehensive measure of shifts in DOM composition, i.e. capturing shifts in DOM composition that cannot be attributed solely to either PC1 or PC2.

The model for dispersion was driven by high flow in the wet winter and spring months (November, December, January Flow, and March, April, May Flow, Fig. 8a) and low flow in the dry summer months (June, July, August, September Flows, Fig. 8a). Standard deviations of these flow magnitude indices showed a similar trend of positive correlation in wet months and negative correlation in dry months. Moreover, the dispersion increased with increasing magnitude and frequency of short duration high flow events (1-3-7 Day Maximum and 1-3-7 Day Frequency, Fig.8a). Regarding the rate and frequency of flow change, the count of increasing flow days decreased and the count of decreasing flow days increased this dispersion (No Increasing Flow Days, No Decreasing Flow Days, Fig. 8a). Regarding the timing indices, the later date of the annual minimum flow was related to higher dispersion whereas variation in this date (i.e. its standard deviation) was related to lower dispersion.

In addition to identifying the drivers of dispersion, we observed that PC2 variance decreased with increasing long-duration minimum flows specifically, while PC1 variance increased with increasing high flood frequency and counts. Similarly, PC1 variance increased with increasing rise rate and decreased with increasing fall rate. Additionally, we found that PC1 variance decreased with increasing SD of minimum flood-related indices especially, while most SD indices were negatively correlated.

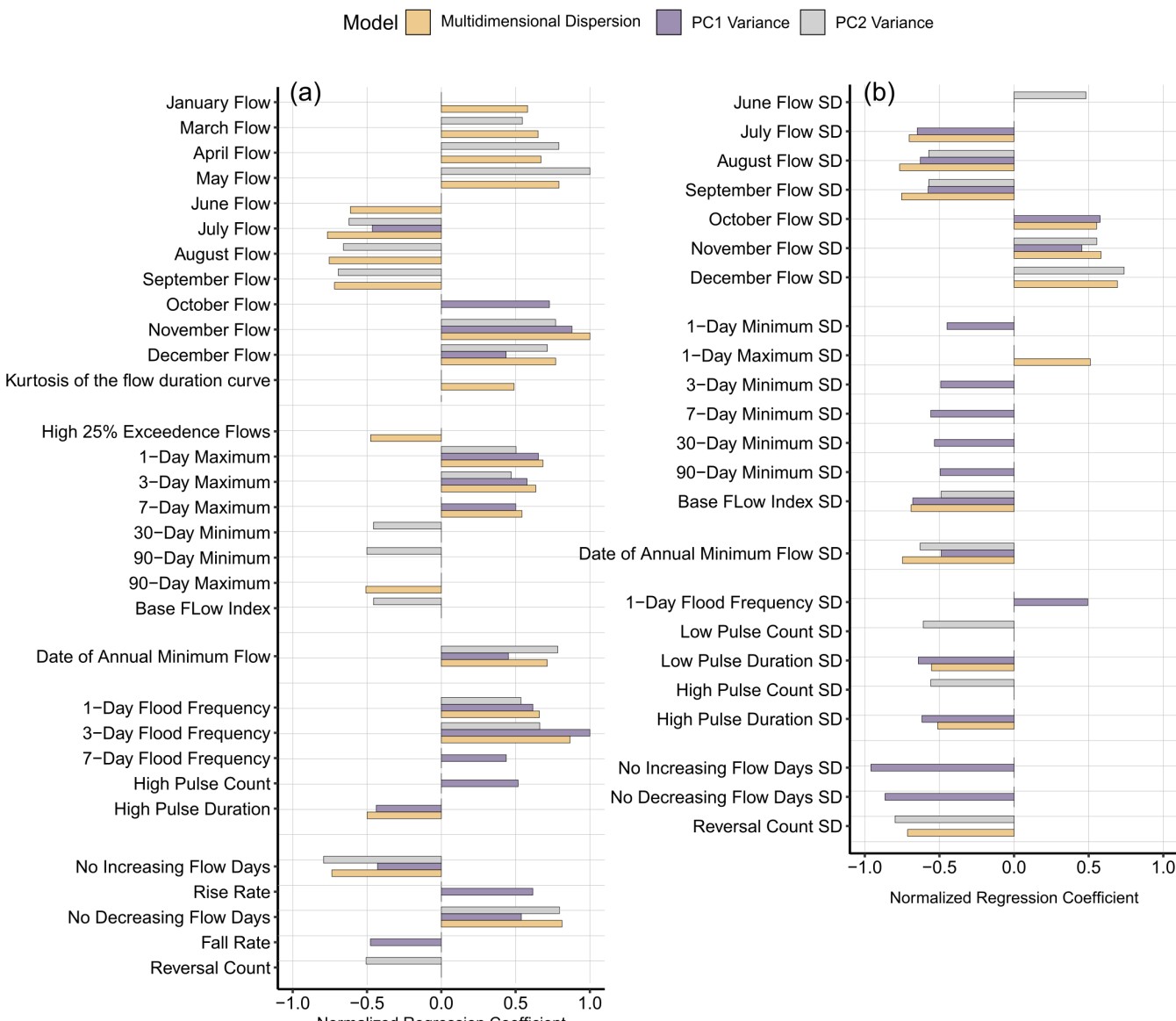

**Figure 8.** Following the PLSR model the indices with VIP scores higher than 1 were assessed for each model. (a) The mean indices and (b) the intra-annual standard deviation indices are separated for ease of interpretation. The yellow bars represent normalized regression coefficients for the indices in the model explaining multidimensional dispersion, the purple and grey bars represent the regression coefficients for the indices in the models explaining variance along PC1 (ranging from -0.07 to 0.07) and along PC2 (ranging from -0.06 to 0.06), respectively. For better comparison, regression coefficients were normalized to ranges [-0.02, 0.02], [-0.07, 0.07] and [-0.06, 0.06] for dispersion, PC1 and PC2, respectively. The indices that belong to the same sub-group are grouped together. The complete list of indices used and their abbreviations are given in Supp Table 3.

## 4 Discussion

In this study of 20 rivers, we compared the effects of dams altering two distinct natural flow regimes on DOM regimes. The literature suggests that dams do not influence all rivers alike (McManamay et al., 2012), and indeed, the likely high diversity of DOM regimes associated with natural flow regimes makes similar reactions to damming appear improbable. On the other hand, a suite of common ecological effects associated with reservoirs, e.g. the simple fact of increased residence time, may indeed drive commonalities of DOM regime response to a dam-induced alteration of natural flow dynamics. In reality, the natural flow

regime forms a baseline, on top of which a dam with a reservoir of a certain size and type of operation, may alter flow with consequences for the DOM regime. We acknowledged this in our study design by restricting study rivers to two distinct natural reference flow regimes, namely Atlantic and Mediterranean.

Notably, in several aspects, DOM regimes were more variable across the natural and altered Mediterranean rivers, hampering the identification of similar responses to dam-induced flow alteration in Atlantic and Mediterranean regimes by a strict compar-

ison of altered rivers to their natural equivalents. However, DOM regimes showed consistent features in both altered Atlantic and Mediterranean rivers. We suggest that reservoir effects (i.e. hydraulic residence times and operation schemes (Maavara et al., 2020)), which cause flow seasonality shifts and flow regime smoothing, may work towards altered DOM regimes in downstream systems along a range of pathways. This may ultimately lead to unique outcomes for each river, yet with some consistencies despite rivers belonging to different reference flow regimes.

### 4.1 Investigated flow regimes and observed flow alterations

This study benefits from Peñas and Barquín (2019) who classified natural flow regimes on the Iberian Peninsula into 20 distinct regime types (or hydrological classes), of which we selected two. Indeed, natural rivers in both classes have pretty distinct flow regimes, each with common features. Hence, both natural flow regimes have a low variance of flow descriptors among their rivers (Supp Fig 1c). Contrarily, the high among-river variations of flow variability boxplots in most altered regimes (Fig. 4),

combined with the hydrographs (Fig. 3), indicate little commonality in flow regimes across the altered rivers: while some rivers have smoother hydrographs, others show a tendency towards increased temporal variation in their flow regime (e.g. introducing summer high flows and/or winter low flows) or show little influence of the dam on the seasonal time scale studied. This is in accordance with the literature suggesting diverse effects upon damming, and while it may push some rivers outside of their normal hydrological functioning (McManamay et al., 2012), others may still behave similarly to their expected natural flow

regime.

As visible by their higher among-river variation, flow alterations diversified Atlantic rivers more than Mediterranean rivers (Supp Fig. 3). Indeed, both reference flow regimes diversified into different directions, mostly due to various reservoir designs, purposes and sizes. Altered flow regimes are thus not as easily classified and not as distinct as their natural equivalents. However, we also observed two common features of altered flow regimes: (1) the general loss of dynamics throughout the year by

less intense 1-Day High Flows (Fig. 4a) and more intense seasonal low flows (Fig. 4e-f), and (2) the newly introduced summer high flows resulting in a shift of flow seasonality (Fig. 4 b and e). The latter was especially pronounced in Mediterranean rivers,

where reservoirs exclusively serve the purpose of irrigation and thus release water to arid downstream regions during summer. Additionally, the reservoir capacity is surpassed in some but not all smaller reservoirs in late winter to early spring, resulting in an unreliably appearing secondary earlier "release", hence the double-humped shape of some altered flow regimes. Reservoirs of our study ranged in size between 2 hm$^3$ and 664 hm$^3$, thus some but not all were big enough to withhold high spring flows for storage until release in a single summer peak. A similar seasonal water release does not exist in the altered Atlantic rivers, especially those flowing to the North (Nalon, Nanca, Narcea). These hydropower reservoirs have low storage capacity, hence - at the studied seasonal time scale - they more closely mimic the intense summer low flows of their natural equivalents. We note that hydropower as a purpose does not principally prevent seasonal water storage for later hydropeaking (Almeida et al., 2020), yet hydropeaking was not observed in the studied rivers.

Thus, in our study, flow regimes diversify through common, yet varied types of alteration. Our study design implied the creation of inherently heterogeneous "altered" flow regime groups with reservoirs differing in size and residence time, but also facilitating various stratification patterns and showing various warming potentials. One thing our study cannot achieve by design is to look for commonalities in DOM regimes for reasonably categorized, distinct types of flow alteration. In fact, grouping reservoirs by purpose (irrigation vs. hydropower) resulted in mostly little difference between the studied DOM regimes despite their distinct flow features. Thus, we believe any commonality we find based on a coarser definition of mere "dam-induced flow alteration" may be of prominent importance and strong applicability.

### 4.2 Responses of annually averaged DOM quantity and composition towards flow regimes

In our comparison of annual averages of DOM quantity (DOC) of natural and altered rivers, tests missed significance likely due to low statistical power despite median values of altered rivers appearing higher specifically in the Atlantic rivers. Yet, a clear-cut response of DOM quantity to flow regimes or their alteration is maybe naive to expect. DOM quantity of natural rivers is affected by various environmental factors such as topography, climate and land cover. In contrast, in altered rivers, it is additionally influenced by factors such as dam operation, reservoir stratification and reservoir water temperature (Maavara et al., 2020). Various studies have found conflicting results on reservoirs as increasing, decreasing or not altering DOC concentrations for the downstream systems (Ulseth and Hall, 2015). In rivers from arid regions with high flow variation for example, upstream reservoirs may act as a carbon source (Parks and Baker, 1997) or a carbon sink (Miller, 2012). In temperate regions, upstream reservoirs may act as a carbon source (Knoll et al., 2013), while no change was observed in reservoirs of boreal rivers (Nadon et al., 2015; Imtiazy et al., 2024).

The existing diversity of influences makes a common response of DOM quantity to flow alteration across Atlantic and Mediterranean rivers unlikely, especially when considering the highly integrative nature of a variable like annually averaged DOC that integrates DOM across various sources and chemical compositions as well as over time. Yet, in our study, we observed an increasing trend in DOC from natural to altered rivers when Atlantic and Mediterranean rivers were separated. One explanation for this apparent effect of dams on DOC concentration could reflect a sampling problem associated with the efficient capture, transient storage and averaging of extreme DOM amounts delivered to the river network from the catchment during short-term flood events (Raymond et al., 2016). In this explanation, dams with large enough reservoirs interfere with

the downstream transmission of a DOM pulse during a flood, but more importantly, this DOM pulse is transiently stored in the reservoir and - if not subjected too strongly to bio- and photodegradation - passed on towards downstream later over a longer time. Thanks to this interference of dams with naturally occurring flow and DOM pulses, downstream DOM composition is "hydrologically averaged" and invariant in time. Thus, sampling at discrete points in time in altered, downstream rivers results in relatively high and stable values, while it is highly unlikely that with just six sampling dates we were able to capture a pulse on any of our natural rivers, whose annual mean DOC then ends up underestimated. In addition or alternatively to this effect of temporal smoothing of terrigenous DOM pulses, productivity in reservoirs may contribute additional DOM of autochthonous origin, which could potentially also counteract losses by bio- and photodegradation. All these mechanisms are associated with residence time of water in the reservoirs, yet they expectedly result in different DOM compositions passed on to downstream river reaches.

Strictly looking at annual average measures for DOM composition between natural and altered flow regimes we found only limited evidence for differences; we observed a significant flow alteration effect on mean DOM composition only along PC2 for Atlantic rivers. Here, altered rivers appeared on average more humic with signs of photodegradation (SR). The (further) lack of significant effects between naturally flowing and flow-altered rivers with regard to (average) DOM composition to some extent supports the idea of reservoirs as smoothers of a variable, natural DOM regime, or it may imply that other factors such as hydrology or catchment characteristics have an overriding influence on DOM composition, thereby masking any potential reservoir effect by a lot of noise.

In fact, the literature on linking DOM composition to dams has mixed results. Some studies show that a high export of autochthonous and biolabile DOM caused by increased primary production in reservoirs and tailwaters may result in a shift from terrestrially sourced DOM to protein-like DOM downstream of dams (Imtiazy et al., 2024; Ulseth and Hall, 2015; Nadon et al., 2015). Contrarily, other studies claim that dams of certain size and function may have no significant effect on downstream DOM composition. In boreal rivers, for example, no shift in DOM composition from terrestrial to protein-like was observed downstream of hydropower and storage dams (Nadon et al., 2015). However, another study from the same climatic region observed that increased residence time in a hydropower-producing mesotrophic reservoir shifted the DOM composition from allochthonous to autochthonous (Imtiazy et al., 2024). This diversity of responses again points to a particular source of noise in our study that may cause difficulties in finding clear-cut effects of flow alteration on DOM composition: the fact that we grouped various dams of different purposes and sizes under the term "flow alteration". In this respect, reservoir trophic state and dam operations such as the timing of the reservoir filling or water release are also of concern, specifically regarding their influence on the composition of DOM transported to downstream reaches by interfering with residence times and transmission of upstream source signals (Xenopoulos et al., 2021). Indeed, dams can be a carbon source or a carbon sink depending on many such factors related to dam nature and operation; potentially, they cause DOM composition to respond to "flow alteration" in opposing directions.

### 4.3 DOM regime reactions: Atlantic and Mediterranean rivers show divergent responses to flow regime alterations but with similar outcome

A DOM regime may be better described than just by annually averaged DOC and DOM composition, especially when these variables are subject to dynamic processes and show pronounced temporal variability. Indeed, when looking at the temporal change of DOM composition indicators (Supp. Fig. S2), we observed specific differences in summer and fall months between the natural and altered regimes in both Atlantic and Mediterranean rivers. The divergence between natural and altered rivers is even more obvious in the multivariate analysis, specifically along PC2 as well as in $\beta/\alpha$, where particularly during summer,

DOM in altered rivers behaved differently than in natural rivers (Supp. Fig. S2c-f).

To make use of the temporally spaced information without having to resort to time series analyses, we computed measures of variation of DOC and of shifts in DOM composition at the annual timescale as descriptors of the DOM regime. We reasoned that such integrative measures based on variance can also be quite flexibly computed in cases where the temporal spacing of measurements is uneven, not aligned among rivers, or sampling is sparse or spread randomly over multiple years.

Hydrological seasonality and event-induced flow variation are the main drivers of annual variation in carbon concentration in natural rivers (Fasching et al., 2016). However, we could not observe an effect of flow regime on the annual DOC variation nor observe any clear difference among annual timelines of DOC (Fig. 5b-c). Contrarily, we found several DOM indices showing differences in their annual variance (CV values) and in multivariate dispersion among natural and altered flow regimes. Specifically dam-impacted Atlantic rivers showed consistently lower temporal variation of various indices, multivariate dispersion

and PCA-axes. Thus, dams in these rivers led to a significant temporal homogenization of DOM in downstream sections. These patterns appeared in Mediterranean rivers as well, most notably when assessing multivariate dispersion (Fig. 6), However, in Mediterranean rivers, the differences between natural and altered flow regimes were insignificant for most indicators, largely because of remarkably higher spatial variation of temporal CVs across the various natural Mediterranean rivers. DOM regimes in naturally flowing Mediterranean rivers were indeed either quite variable over time or fairly stable - a finding that strongly

contrasts the consistently dynamic DOM regimes of natural Atlantic rivers (Fig. 6, 7). Taken together, our results suggest that dams may lead to fairly consistent and temporally stable DOM regimes in downstream river sections even in climatically different reference flow regimes and contrasting the high diversity of DOM regimes potentially appearing in natural flow especially in Mediterranean rivers.

### 4.4 Dam and reservoir-associated mechanisms behind DOM regime reactions to flow alteration

Our PLSR results linking flow regime characteristics to multidimensional dispersion and variance along PC1 and PC2 axes yield more insights into the mechanisms behind seasonal shifts in DOM composition. Short-duration high-flow events are key in mobilizing terrestrially stored carbon into the riverine system via increased riparian connectivity. The PLSR results show wider annual shifts in DOM composition with an increase of indices describing magnitude (1-3-7 Day Maximum) and frequency (1-3-7 Day Flood Frequency) of such short-duration high flow events (Fig. 8a). These relationships may be partly

driven by natural differences in terrestrial DOM mobilization, but likely they are also associated with the ability of reservoirs to interfere with the downstream transmission of a terrigenous DOM pulse.

Indeed, many reservoirs in Northern Spain are associated with a decreased frequency of high flow events due to the storage of water in the reservoir and downstream flow smoothing (Aristi et al., 2014). In our study, in both natural flow regimes, summer low flows follow high winter and spring flows (with some climate-specific variation, for instance in the exact timing and intensity of drought). Thus, flow alteration largely leads to a more or less strong reversal of seasonal dynamics by retaining
water in reservoirs in winter and spring for later release during summer. This effect is greater in Mediterranean rivers. This seasonality of the flow regimes in our study is reflected in patterns of shifting DOM composition: PLSR modelling identified a positive effect of mean monthly flows in winter and spring (floods) but a negative effect of mean monthly flows in summer (droughts) on the magnitude of shifts in DOM composition. This is corroborated by relatively higher terrestrial signals in
the rainy spring months in natural rivers (Supp. Fig. S2a-b) and also shown by the positive effect of seasonal high flows (March, April, May) on PC1 indicating aromaticity and molecular weight variance. This is in accordance with the current scientific consensus that high flows are generally linked to allochthonous DOM inputs while low flows promote autochthonous production (Imtiazy et al., 2024). Natural flow regimes allow the transmission of flood-induced terrigenous DOM pulses in spring while such DOM is rare in these rivers during summer low flows, when there is increased opportunity for production
of autochthonous DOM. On the other hand, in altered flow regimes, reservoirs pass on seasonally smoothed and thus rather invariable terrigenous DOM to downstream systems long into summer (Supp. Fig. S2e-f). The negative effect of summer flows on the magnitude of shifts in DOM composition is likely as much driven by low summer flows in natural flow regimes as by high summer flows in altered flow regimes. In altered flow below dams, the lack of summer drought lowers the natural variation of DOM composition of a river.

On the other hand, low flow during naturally dry summer months may directly increase the variation of DOM composition through higher microbial signals from autochthonous production of DOM in the merely flowing rivers. This is shown by higher PC1 and PC2 values during summer low flow in the natural Atlantic rivers while DOM remained comparably invariable in the altered Atlantic rivers (Supp. Fig. S2). A similar microbially sourced carbon pulse was observable in altered rivers in October, coinciding with the lowest flow rate for these rivers. This finding still agrees with the earlier interpretation about reservoirs
as hydrological smoothers of upstream DOM signals, which, however, may include not only terrigenous DOM but also in-stream produced material. Within reservoirs, these two types of material may then be subjected to photo- and bio-degradation, ultimately leaving less labile humic material for continuous downstream export with little compositional variation.

A counter-effect to the lowered autochthonous contribution from upstream rivers to DOM variance in the altered rivers may be the potential production of DOM in the reservoirs, especially when reservoirs have longer residence times (Ulseth and Hall,
2015; Imtiazy et al., 2024), reasonable nutrient (Maavara et al., 2020) supply and support warming. For Atlantic rivers, our data do not suggest reservoirs as sources of new autochthonous material. For Mediterranean rivers, summertime excursions of DOM towards higher PC1 scores (i.e. towards more microbial DOM, higher FI and $\beta/\alpha$ ratios, Supp. Fig. S2 bdf) suggest some influence of DOM production within reservoirs, even though this effect could not be identified as statistically significant. These results agree with the generally good ecological status and thus low trophic state of the reservoirs (Table 1).

Notably, the buffering effect of reservoirs may also imply a change of DOM composition due to higher degradation as a result of longer residence time. We observed that in August - October measurements, PC2, which is strongly correlated to SR and E2:E3, diverges between natural and altered rivers in both Atlantic and Mediterranean regimes. Here, natural rivers have positive values and altered rivers have negative values. We think this may be related to the higher irradiation as SR indicates photodegradation. This might be due to two factors; longer residence times in the reservoir leading up to these moments of
sampling and water release from the reservoirs over summer that decreases water depth, hence increasing light penetration.

The pervasive effects of dams on DOM regimes also had an interesting additional effect when looking at the spatial variation of DOM regimes across rivers. When assessing DOM composition, we observed that flow alterations homogenized the DOM regime across rivers despite our dams being of differing sizes and purposes, altering flow regimes in various directions (e.g., Fig. 6). This loss of natural spatial diversity of DOM regimes across rivers indicates that dams have a powerful homogenizing
effect on DOM regimes at the landscape scale. Interestingly, we could not find such an effect for variation of DOC and $CV_{DOC}$. Here, data inspection (Fig. 5, Table 4) shows a non-significant trend towards increasing spatial variation across rivers from natural to altered rivers. This would indicate a spatial diversification of DOM regimes in terms of quantitative behaviour, which parallels the flow regime diversification due to dams.

## 5    Conclusions

In our study, we investigated how dams, serving various purposes, alter flow and DOM regimes in rivers with two distinct natural (reference) flow patterns. We concluded that dams in our study can move rivers outside of their normal river function both in terms of annual average and annual variance of DOC concentration and composition towards surprisingly similar outcomes despite differences in underlying natural hydrology. Dam effects such as high flows during natural seasonal drought and interference with natural high and low flow events are likely the driving factors for the decrease in seasonal shifts of DOM
composition. A main mechanism is the prevention of transmission of upstream-sourced terrigenous and autochthonous DOM towards downstream-located systems. In conclusion, the often drastic seasonality reversals of the flow regime can strongly impact the seasonality of the DOM composition. It appears that in our study area, reservoirs act as a buffer, where longer residence times average out the naturally occurring shifts in DOM composition by trapping the upstream-sourced carbon inputs in the reservoir for a longer time and objecting them to bio and photo-degradation, thus sending relatively invariable
DOM further downstream. This effect was similar across all altered flow regimes despite the diversity of dam types and purposes.

Flow alterations thus resulted in the homogenization of DOM downstream of dams across time (within each river) and space (across rivers). This may have important consequences for riverine carbon cycling. In natural flow regimes, considerably variable DOM meets benthic bacterial consumers that undergo frequent succession due to disturbance by flow. The likely
resulting mismatch between the chemical traits of incoming DOM and the microbial functions needed to process this DOM as a resource limits its metabolism (Talluto et al., 2024). In contrast, in altered flow regimes, there is an increased chance for a match between diverse but stable carbon resources and functions of their heterotrophic consumers in now less disturbed

microbial communities. Ultimately, this would translate to increased metabolism of terrigenous DOM and increasingly higher $CO_2$ emissions from rivers downstream of dams. Notably, this yet-to-be-tested effect on riverine carbon cycling comes on top

of already-known effects on intensified DOM processing caused by higher residence times in reservoirs. Increasing knowledge on this frontier could also help to design flow alteration strategies that prevent impacts on ecosystem structure and functioning and move one step forward in restoring natural riverine habitats.

*Code availability.* The R code used for data analysis and visualisations in this paper is openly available on Zenodo at zenodo.14992677 (Kubilay, 2024b)

*Data availability.* The optical data used in the analysis is openly available on Zenado at zenodo.13354316 (Kubilay, 2024a). Hydrological data is obtained from Pompeu et al. (2022) and Goldenberg-Vilar et al. (2022) and can be accessed through the authors.

*Author contributions.* SK: Conceptualization, Methodology, Software, Formal Analyses, Visualization, Writing – Original Draft and Review and Editing. EE: Conceptualization, Data Curation. JBO: Funding acquisition, Conceptualization, Writing – review & editing. GS: Funding acquisition, Conceptualization, Methodology, Supervision, Writing – review & editing. All authors have read and agreed to the published

version of the paper

*Competing interests.* Some authors are members of the editorial board of the journal Biogeosciences

*Acknowledgements.* We thank our colleagues from IHCantabria for their help in sample collection. We also thank our colleagues from Leibniz Institute of Freshwater Ecology and Inland Fisheries for sample processing. This project has received funding from the European Union's Horizon 2020 research and innovation programme under the Marie Skłodowska-Curie grant agreement No 765553.

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
