# Peer review of "Responses of riverine dissolved organic matter to damming in two distinct hydrological regimes of Northern Spain"

_EGUsphere, 2024_

## Author Comment (AC1)

**Anonymous Reviewer #1:**

**Review report**

Title: Riverine dissolved organic matter responds to alterations differently in two distinct hydrological regimes from Northern Spain

This study examines the effects of anthropogenic flow alterations, primarily caused by dams, on DOM concentration and composition in Spanish rivers of the Atlantic and Mediterranean region. This research compares rivers with natural and altered flow regimes and looks at how different flow components impact the DOM regime, such that altered Atlantic rivers generally show lower DOM composition shifts compared to natural ones, while Mediterranean rivers appear more resistant to flow alterations, maintaining relatively consistent DOM characteristics.

The study is overall well conducted, relies on a sound empirical basis and uses advanced statistics to identify patterns. The authors introduce the topical background excellently. In that sense I think this is definitely publishable and interesting to the EGU readership. However, there are several issues that I think need some close attention to increase the accessibility and clarity of the study. There are, in my opinion, terminology and reasoning aspects that needs improvement. I hope my suggestions in this regard are helpful.

**REPLY:** We thank the reviewer for their positive evaluation of our work.

**General comments**

Regarding the study concept and abstract, and even for someone who works with DOC, the goals and findings of the study are not easy to grasp. I think this has partly to do with the comprehensive aspiration: the authors do not only want to look into DOM "regime" shifts after flow alterations, but also compare these shifts in two different river system types, and seek for the system properties that are statistically connected to response. This is tough to comprehend, and it does not help that the terminology is at times imprecise and self-defined: DOM "Turnover" is used here differently than in most other contexts (where it essentially means transformation and/or mineralization) – is "compositional shifts" not clearer? I also have problems to understand was is meant by "annual DOM composition" (L12), "temporal turnover indicators" (L256) and several other derivatives of the DOM-related language. I suggest to revisit the part of the work that introduces the terminology use in general, and specifically the analysis goals, concepts and expectations, and harmonize the language related to these. One headline in the results "Linking DOM regimes to flow regimes" could for example be used more often.

**REPLY:** Thank you for pointing this out. We agree that some of the terminology used in the manuscript, particularly the term "turnover," may be confusing. While we originally borrowed "turnover" from community ecology, where it precisely means "shifts in composition", we acknowledge that it carries an established meaning in other contexts, which can lead to misinterpretation. To improve clarity, we will replace this term with less ambiguous alternatives, such as "compositional shifts," as suggested.

Similarly, we recognize that expressions like "annual DOM composition" are imprecise. We propose revising this term to "annual average DOM composition" to better convey our intended meaning. Additionally, we will carefully review the manuscript to harmonize the terminology with the study's goals and concepts.

To further enhance clarity, we will provide detailed explanations of key terms in concert with an explanation of the study concept in a conceptual graph presented early in the paper. We believe these measures will improve the manuscript's readability and address your concerns regarding the difficulty in grasping the study's goals and findings.

**Specific comments**

L30-32: "This highly reactive fraction…" a reference is needed.

**REPLY:** Here, we will add the necessary citations.

L34 but also temporally (Catalán et al., 2016)… not an adequate citation in that context, because that work really looks at spatial differences of a time-reated property

**REPLY:** Thank you for noticing. We will add the necessary citations.

L53 This rather general model of a DOM regime´s reaction to damming needs fine-tuning… quite a colloquial language for the central part of the study motivation

**REPLY:** We will change the language here.

L55 inflowing DOM concentration: not really the concentration but the amount

**REPLY:** We will correct this.

L56 I don't agree that "all" these biotic factors are "associated" with the natural flow regime

**REPLY:** True. We will update this sentence.

L 58, the term "compositional turnover of DOM" needs to be clearly defined, see above general comment.

**REPLY:** We will change this wording (see our answer above) and clearly define what we mean by it here where it first appears.

L63 two naturally defined hydrological classes,.. this is the first appearance outside of the Abstract and the relevance of this concept demands appropriate introduction on first appearance

**REPLY:** Thank you for noticing, we will add a definition before this point.

L65 We expect the effect of flow regime alterations on the DOM regime to depend on certain characteristics of the natural flow regime. … this is an unintuitive research goal, what "characteristics" could this be?

**REPLY:** We apologise for the confusion, this was not well formulated. In fact, we here merely wanted to suggest that effects of flow regime alterations on the DOM regime will depend on the initially unaltered flow regime. Natural flow regimes (and likely also associated DOM regimes) are quite diverse. And even if flow alterations are also diverse, any alteration of the DOM regime still happens from a certain baseline dictated by the natural flow regime. We will reformulate this part of our research objectives and support our reasoning with a conceptual figure laying out our hypothesis framework. In this, we will also point to our analysis in the last section of the study where we try to identify hydrological drivers of DOM regime characteristics among a large number of hydrological indices.

L 161-163, the authors state that the sampling dates to the centroid of a river serves as a measure of temporal turnover of DOM and it is computed as a dispersion. There is not a clear explanation of what this dispersion precicely means and how it is derived. More explanation would be useful.

**REPLY:** We will replace the phrase "temporal turnover" by the less ambiguous phrase "temporal shifts in composition" and will explain better how it can be captured by a multivariate measure of variation (i.e. "dispersion").

L259, what are the "temporal turnover indicators"? These indicators are not explicitly defined.

**REPLY:** These will become "indicators of temporal shifts of DOM composition". We will add a clear description in the methods to explain the coefficient of variation calculations and how they serve as such indicators.

L309 blurry but more encompassing… not sure I understand what you mean here

**REPLY:** Our intention is to express that a multivariate measure of temporal compositional change of DOM ("turnover") like "dispersion", which is influenced by all included DOM indicators, is a very integrative measure, yet this comes at the cost of less precise meaning. We will update this sentence and use less colloquial language.

L 361, "hydropeaking" may need definition or referencing

**REPLY:** We will add a reference here.

L477-479, sentence quite long, consider dividing

**REPLY:** We will divide this into 2 sentences.

Fig 2 add aM irrigation

**REPLY:** Thank you for noticing, we will add this to the legend.

Table 4 sample n and frequency may be a useful information here

**REPLY:** We will add sample sizes and the number of sampling occasions to the legend of the table.

Figure 7 flow properties: these should be introduced at one point earlier in the text. Maybe revise the image altogether because it is hard to read and unexpectedly complex for this stage of the manuscript. Why not for example instead of grouping by category, sort by influence, or withdraw from showing *all* influences and select the most significant/important ones. I believe this would increase the interaction with the information massively.

**REPLY:** Indeed this figure is very complex and we have tried many alternatives for the sake of saving space and increased clarity. We believe dividing it even more would create more confusion than provide clearance at this point. The figure already shows only a subset of all the indices used in the model, selected according to VIP values. Also, we wish to keep the aspect of comparability among the 3 models built for the three response variables, as this makes the plot both informative as well as space-efficient. The current plot version also nicely shows that there is a trend throughout the year in flow magnitudes (positive correlation in winter, negative in summer) as well as in other indices such as minimum-maximum daily to monthly events and other contrasting indices of high water vs low. We will think of some additional graphical ways to improve the interaction with the information.

494 subscript $CO_2$

**REPLY:** We will correct this.

Citations of the Xenopoulos review show up several times. Maybe it is useful to cite original study in some cases?

**REPLY:** We will update these references to the original study where applicable.

We thank the reviewer for their insightful and constructive comments.

---

## Author Comment (AC2)

Unexpectedly, two appointed referees could not provide review reports even after the extended due date. Therefore, Associate Editor provides another required report to expedite the delayed review process. – Ji-Hyung Park

**General comments**

This study investigates the effects of natural and altered flow regimes on DOM concentration and composition in several rivers across Northern Spain. It is a well-designed study employing a combination of analytical tools and advanced statistical approaches. The key findings on the complex linkages between flow regimes and DOM are novel and intriguing. Although the manuscript is generally well written, it also requires both clarifications of several uncertainties and improvements for a more focused discussion as follows:

**REPLY:** We thank Ji-Hyung Park for the positive evaluation of our work.

1. Hypotheses and key themes: This study examines the effects of altered flow regimes under two different natural settings. As the authors stated ("We hypothesize that DOM properties respond to both natural flow regimes and flow alterations. We expect the effect of flow regime alterations on the DOM regime to depend on certain characteristics of the natural flow regime."), the concentration and composition of DOM would respond to changes in natural flow characteristics without any anthropogenic perturbations to flow regimes. Therefore, the quite general hypothesis needs to be more articulated with regard to which specific DOM characteristics would respond to which specific characteristics of flow regimes (both natural and anthropogenic). Furthermore, the dams in the studied region appear located in headwaters or upper reaches. Given the fact that flow regime changes vary with dam types and locations, dam effects, as a key theme of this study, need to be explained for the prevailing dam characteristics of the study region.

**REPLY:** Thank you for your comments. We will revise our hypotheses to better articulate the expected DOM regimes and their expected changes due to flow alteration under the various river conditions.

In this study, our primary focus was on examining DOM regime changes under different flow regimes, in the case of 'flow regime alterations' we aimed to achieve this, irrespective of dam types or purposes. Therefore we grouped various dams under the general category of "dam effect" within the same underlying natural flow regime. A strength of the study is that it is built on a previous study allowing us to define a "natural" flow regime type even if a dam is already in place. This enables us to identify overarching trends in the DOM regime caused by dams while simultaneously acknowledging that specific regional climates likely shape flow regimes while also influence the design and operation of reservoirs..

**REPLY:** Despite this general overall goal, we recognize the necessity to delve deeper into the prevailing dam features of our study. We have discussed the "dam characteristics and operations" relevant to altered flow regimes in Section 3.1, where we connected these characteristics to the water demands of the region. We will

expand on the common features of the dams in our study, such as their locations (e.g., predominantly in headwaters or upper reaches), the absence of hydropeaking, and their storage capacities. These details will be incorporated into the Methods section to offer a clearer context for the observed dam effects without going into the details of a site-based analysis.

2. River classification: The authors describe as if they studied 20 different rivers, but actually Fig. 1 shows 20 river locations. Some of them, as tributaries, belong to the same river basins. Around five basins are delineated on Fig. 1. Please articulate how many river basins and their tributaries were studied; and use relevant terms to distinguish mainstems and tributaries. In the same context, the river classifications in L 78-81 need revision. Some of the river sites belonging to two different systems (Atlantic and Mediterranean) are shown shoulder to shoulder on Fig. 1, making me wonder whether these sites really belong to two different "climatic regions". For instance, look at site Arlanzon. From the names, I expected that these rivers might discharge into either the Atlantic Ocean or the Mediterranean Sea. In my view, the use of these names needs to be confined to indicate something like "Atlantic-type flow regime" after defining it at its first use.

REPLY: Thank you for pointing this out. The layout and terminology we used indeed reflect some complexities, partly due to reliance on a previous study that was used to constrain our study design. In the original study by Peñas et al. (2020), Spanish river systems were classified based solely on hydrological indices, without considering their climatic region or geographical location. Subsequently, the names "Atlantic" and "Mediterranean" were assigned to two of the thereby defined hydrological groups based on their predominant flow regime characteristics: Mediterranean rivers typically experience long dry summers, while Atlantic rivers exhibit shorter summers with more moderate flow year-round. We recognize that this naming convention might not be intuitive at first glance, as it is based on hydrological patterns rather than geographic location or basin.

To address your concerns, we will clarify in the manuscript that these classifications represent "Atlantic-type" and "Mediterranean-type" flow regimes, and they are not indicative of the location of the basins they belong to.

Regarding your observation about tributaries, it is correct that some of the studied "rivers" are located within the same river basin, i.e. they share the same downstream located mainstem. However, we ensured that no study site´s catchment (and thus source of water and DOM) is nested within another one´s to maintain spatial independence. We will update the methods section to specify the number of basins and rivers included in the study and to clarify spatial independence.

3. Data interpretation: According to Table 4, DOC concentrations are significantly different only between nA and Am. Associated descriptions in Abstract and Results (L 226-230) need to be consistent with the presented results. Although the means were not significantly different, inter-regime differences distinct for some of the river sites (Fig. 4) warrant descriptions in L 226-230, particularly in relation to (any common) specific hydrologic characteristics of these sites. Statistical analyses indicated the dominant role of high-flow periods (L 311-313) in explaining variations in DOM composition. Can this be related to flow-controlled variations in DOC conc.?

Regarding the lack of dam effects on DOM composition indices, please refer to specific comments below.

**REPLY:** Thank you for your comments. You are correct that the DOC concentrations between nA and Am are different, as also reported in Table 4. As our primary focus was to compare rivers within the same regime or to compare the two natural regimes to emphasize their distinct characteristics, we did not want to stress this (less interesting) result. That said, for clarity and consistency, we will extend the text descriptions of these statistical results to all possible pairwise comparisons. We will also further explore some of the more distinct DOC or DOM regimes, which more prominently emerge beyond the pre-defined flow regimes, in relation to hydrological features of the respective sites (or sets of sites).

Specific comments

- Title: Do you mean that…two distinct hydrological regimes "of" Northern Spain?

**REPLY:** Thank you for noticing this, indeed "of" Northern Spain would fit better. We will update this.

- Line (L) 2: What have altered flow regimes?

**REPLY:** Many rivers specifically from this region have altered flow regimes, we will update it to "of this region" in this sentence.

- L 3 and thereafter: Please remove the comma after the subject and check grammatical errors and typos throughout the manuscript including tables and figures (for instance, superscripts such as m2 and concentration units in parentheses).

**REPLY:** Thank you for noticing, we will carefully scan the manuscript and correct such errors.

- L 6 "hydrological classes": As suggested by the first reviewer, please use terms consistently; in this case "flow regimes"? Definitions are required for the key terms including flow regimes, DOM regimes, and DOM turnover.

**REPLY:** We agree that these terms create confusion at various points in the manuscripts. We will clearly identify a set of terms early in the manuscript and also in a conceptual graph, and stick to those throughout the text. Instead of "DOM turnover" we will use "compositional shifts of DOM" in order to avoid ambiguity.

- L 8: Please do not divide this relatively short abstract into two paragraphs.

**REPLY:** We will combine these two paragraphs into one.

- L 12: Was this turnover rate calculated per unit carbon? Higher DOC concentrations might have led to higher turnover rates, so this needs to be clarified.

**REPLY:** The word "turnover" here refers to temporal changes in DOM composition. As we understood from the first reviewer's comment, our usage of the term "DOM

turnover" resulted in some confusion due to the - in the field more common - interpretation of "turnover" as meaning chemical transformation. Hence, we will replace the word "turnover" by "shifts in DOM composition" and clearly indicate its meaning early in the manuscript and in a conceptual graph. With this in mind, we will indicate here clearly that the paper is not looking into "carbon turnover rates" but explores temporal shifts in DOM composition.

- L 14: What are "unusual DOM behavior"?

**REPLY:** We will change this to "deviations from the natural DOM behavior".

- L 19: Please specify "these" in this complex sentence.

**REPLY:** We will break this sentence into two and change "these" to "these natural flows".

- L 53 "DOM regime´s reaction to damming needs fine-tuning": Please rewrite this vague sentence.

**REPLY:** We will re-write this statement.

- L 58 "compositional turnover": Compositional changes, or variations?

**REPLY:** Here we refer to temporal variation of DOM composition, we will update this expression accordingly.

- L 93-94: Please provide more details about the hydroclimatic conditions for these six samplings. Later in Results and Discussion, sampling conditions need to be related to flow regimes and DOM characteristics. It would be helpful if sampling timings are indicated on Fig. 2.

**REPLY:** We will indicate the sampling timings in Fig.2 and in the text here. We judge the idea of indicating detailed hydroclimatic conditions for each site per sampling occasion to require too much space in the text while simultaneously not being very informative as our sites had varying conditions from dry to flooded even within the same sampling campaign. However, to give more insight into site-specific hydroclimatic conditions, we will provide discharge data per site per sampling occasion in the supplementary information.

- L 99-105: Given the importance of DOC quantification and optical characterization in this study, please provide some key QA/QC measures, including blanks, verification standards, references,,,

**REPLY:** We will provide more details about DOC and optical characterization of DOM in the methods section of the manuscript along with the related references. Briefly, we routinely used MQ-water as blanks for DOC measurements and optical measurements (at least one blank run every 10-15 measurements). Also, we had 3 replicates per site and occasion, each measured twice to take into account any instrumental error. To prepare standards for DOC-measurements we used

potassium hydrogen phtalate (KHP). The typical limit of quantification for DOC is 0.5 mg L−1 and the typical analytical precision is 3 % in our TOC-V analyser.

- L 218-211: Please describe in more detail about the meaning of and how to read out the magnitude differences. It is very difficult to follow!

**REPLY:** We will rewrite this text section without abbreviations (or add explanations if abbreviations are unavoidable).

- L 226: It would help understand the temporal variations in DOC conc. if variations in hydrologic conditions among the six samplings were described briefly.

**REPLY:** Per your recommendation, we will add representative sampling dates to Fig.2. We hope this will help to understand these DOC variations as well.

- L 266-267: Interesting approach! The four regimes overlap considerably on Fig. 5a. Do some outlying sites also exhibit high DOC or any common outstanding hydrologic characteristics?

**REPLY:** Indeed, we have looked at whether these extreme sites/occasions coincide with high DOC values but could not find any relation. We will check this more in detail and include our findings here.

- L 307: Again this 'turnover' is an undefined, ambiguous term.

**REPLY:** We will change this term in the final version to "compositional shifts of DOM" as stated above.

- L 324-337: The dams studied in this study appear located in headwater streams. Dams in lowland or high-order streams might be quite different in terms of flow regimes. Please consider this stream-order effects.

**REPLY:** Indeed all dams are in 2nd order or 3rd order streams. We will include stream orders in the above-mentioned table for the supplementary information. We will also point to the fact that our altered flow regime sites were all impacted by headwater dams.

- Discussion is very long and repeating some results descriptions, but limited in citing other relevant studies to put the implications of the key findings in a wider regional or global context. Please consider removing repeated descriptions while focusing discussions on key implications and comparisons with previous studies.

**REPLY:** We will remove the repetitions from the discussion and will strive to improve comparison of our results with those of previous studies.

- L 331 (and other places): Please use two instead of 2.

**REPLY:** We will update this in the manuscript.

- L 351: This very long paragraph can be split here.

**REPLY:** We will divide this paragraph into two.

- L 373-375: Please don't repeat the descriptions already mentioned in Results.

**REPLY:** We will update this sentence not to repeat results.

- L 382: not "increase" but "increasing trend".

**REPLY:** We will update this sentence and use "increasing trend".

- L 387: How can "hydrological averaging" can increase DOC concentrations? DOC flushing mechanisms explained by Raymond et al. assume higher DOC concentrations at terrestrial sources. Increased DOC conc. in stream downstream of reservoirs should then indicate additional supplies from autochthonous sources.

**REPLY:** We wanted to suggest that hydrological averaging by dams that hold back flood water may lead to an increased and stable mean concentration of terrigenous DOM mobilized by floods downstream of the dam. Indeed, this requires a range of conditions, which we did not detail well enough, namely fairly high terrigenous DOM loads mobilized by floods and rather bio- and photo-resistant behaviour. Also, to affect DOC (i.e. DOM quantity), any such terrigenous DOM must still be understood in concert with autochthonous DOM produced in the reservoir. We will improve the description of our idea of "dams flattening a DOM pulse" in the revised manuscript and describe the conditions upon which it may hold. We will do so explicitly here and touch upon it further in the conclusion (per your comment L487)

- L 388-391 & 398-410: Given the significance of dam effects as a key theme, a more in-depth discussion, including comparison with previous studies on DOM characterization in other dam types (not citing only one review paper), is required to explain the observed patterns in aA and aM.

**REPLY:** Thank you for your suggestion. We will include a more detailed discussion of DOM regime effects in light of other studies characterizing DOM downstream of dams.

- L 392-397: Does the lack of statistical differences in DOM composition indices simply translate into no dam effects on DOM? As suggested before, a more detailed look at large within-regime variations is required to figure out some key characteristics driving the link between dam-induced alterations in hydrology and DOM. For example, you are not addressing the trophic status of your reservoirs, which may be quite different from eutrophic reservoirs.

**REPLY:** Indeed, an eventually observed DOM regime downstream of a dam may be driven by a large range of potential influences associated with specific dam features, e.g., location in the continuum or catchment size, reservoir size and residence time, trophic status, and operation regime. Our study and analysis design lumps together dams and reservoirs of various types, thus it was never our intention to identify how specific dam factors may influence DOM. Indeed, we also find that alteration diversifies flow regimes substantially. This diversification of flow regimes may be expected to lead to a similar diversification of DOM regimes, with potential consequence being lower statistical power to identify eventual differences to natural

flow regimes. This may be true when looking at differences in means, yet we let our analysis go far beyond the analysis of "average DOM composition" by looking into potential effects of dams on temporal variation of DOM composition, i.e. a "DOM regime". Surprisingly, we show that DOM regimes do not follow flow regimes as a consequence of alteration; a result of analysing 'variation over time' is that DOM regimes do not diversity but rather homogenize. Thus, we hope to be able to show that simplistic statements like "dams do not affect DOM" are not valid.

Despite our design constraints, one dam-specific factor we identified as potentially important for the DOM regime was reservoir size or storage capacity. Indeed, storage capacity and associated residence time may explain some of the observed among-river differences and we specifically discussed this in section 4.3. We will look into nutrient data and trophic status as potentially interacting with residence time to influence DOM and suggest to discuss potential effects in concert with residence time. Unfortunately, we lack more detailed information about dam operation and prefer to refrain from further speculative interpretation of our data that is not oriented along the principal design of the study. We will look into the possibility of mentioning potential effects of dam storage size in this closing sentence as well.

- L 408: Yes, but which "factors related to dam nature and operation" are critical for your rivers?

REPLY: Please see our answer to the previous comment. Besides generally alluding to a range of dam-specific features that may influence DOM regimes, we will try to identify those in our dataset and link results to particular dams. However, we believe that such an in-depth discussion of particular dams comes dangerously close to speculation and should be limited as it is not supported by our study design.

- L 413-414: Yes, temporality is critical. However, the following descriptions only focus on the statistically analyzed means. As suggested, a brief discussion can be added here to evaluate how hydrologic conditions preceding the six samplings might have (differentially) affected the observed patterns in the four systems.

REPLY: In addition to indicating the sampling dates in Fig 2 as you've suggested, we will add discharge on the sampling day for each site in the SI to give a clear idea on sampling day conditions per site. We will here attempt a limited discussion of how the observed hydrograph of particular rivers may be associated with the observed DOM regime. However, we point to previously already argued limitations of our study design and our limited ability to pinpoint particular dam-specific effects. Currently, we see the need to limit such argumentation based on individual discharge time series to avoid a too speculative discussion.

- L 455-: Please provide some data or literature information to link these trophic changes to autochthonous DOM production.

REPLY: Thank you for noticing, here, we will cite two references, Ulseth and Hall, 2015; Maavara et al., 2020, regarding reservoirs and residence times and reservoirs and nutrients.

- Section 4.4: The title is not in agreement with the text descriptions. This short paragraph is not redundant? Otherwise, provide a more proper discussion.

**REPLY:** This section closes our discussion with a last point looking at among-river variation in low regimes. We will integrate this section into the previous section.

- L 487 "longer residence times average out the naturally high turnover of DOM composition, and send relatively invariable DOM further downstream.": This conclusion cannot explain the observed increasing DOC trend and is not in line with the following implication "increased metabolism of terrigenous DOM and increasingly higher CO2 emissions in rivers downstream of dams". From my own experiences in highly eutrophic, impounded systems, CO2 emissions generally increase with the increasing supply of labile DOM (please refer to this review: https://doi.org/10.1016/j.watres.2022.119362).

**REPLY:** Please see our above explanation of how hydrological averaging by transiently storing flood waters in reservoirs may increase terrigenous DOM downstream of dams. We will improve our explanation of this potential mechanism and the conditions under which it may hold. Our idea of increased metabolism of this DOM is based on an improved match with microbial capabilities. We will improve the explanation of this potential mechanism. Also, we will include opposition to the case of eutrophic reservoirs, which may drive downstream metabolism and CO2 emission by supplying labile DOM.

- Fig. 1: Given the significance of impoundments in distinguishing flow regime types, I wondered if dam locations could be indicated within each altered-regime sub-catchment.

**REPLY:** We will include dam locations for rivers with altered flow regimes in this Figure.

We thank Ji-Hyung Park for their insightful and constructive comments.

---

## Author Response (AR1)

**Anonymous Reviewer #1:**

Review report

Title: Riverine dissolved organic matter responds to alterations differently in two distinct hydrological regimes from Northern Spain

This study examines the effects of anthropogenic flow alterations, primarily caused by dams, on DOM concentration and composition in Spanish rivers of the Atlantic and Mediterranean region. This research compares rivers with natural and altered flow regimes and looks at how different flow components impact the DOM regime, such that altered Atlantic rivers generally show lower DOM composition shifts compared to natural ones, while Mediterranean rivers appear more resistant to flow alterations, maintaining relatively consistent DOM characteristics.

The study is overall well conducted, relies on a sound empirical basis and uses advanced statistics to identify patterns. The authors introduce the topical background excellently. In that sense I think this is definitely publishable and interesting to the EGU readership. However, there are several issues that I think need some close attention to increase the accessibility and clarity of the study. There are, in my opinion, terminology and reasoning aspects that needs improvement. I hope my suggestions in this regard are helpful.

We thank the reviewer for their positive evaluation of our work.

General comments

Regarding the study concept and abstract, and even for someone who works with DOC, the goals and findings of the study are not easy to grasp. I think this has partly to do with the comprehensive aspiration: the authors do not only want to look into DOM "regime" shifts after flow alterations, but also compare these shifts in two different river system types, and seek for the system properties that are statistically connected to response. This is tough to comprehend, and it does not help that the terminology is at times imprecise and self-defined: DOM "Turnover" is used here differently than in most other contexts (where it essentially means transformation and/or mineralization) – is "compositional shifts" not clearer? I also have problems to understand was is meant by "annual DOM composition" (L12), "temporal turnover indicators" (L256) and several other derivatives of the DOM-related language. I suggest to revisit the part of the work that introduces the terminology use in general, and specifically the analysis goals, concepts and expectations, and harmonize the language related to these. One headline in the results "Linking DOM regimes to flow regimes" could for example be used more often.

Thank you for pointing this out. We agree that some of the terminology used in the manuscript, particularly the term "turnover," may have been confusing. While we originally borrowed "turnover" from community ecology, where it precisely means "shifts in composition", we acknowledge that it carries an established meaning in other contexts, which can lead to misinterpretation. To improve clarity, we replaced this term with less ambiguous alternatives, at most places we refer to "compositional shifts" as suggested. Similarly, we recognize that expressions like "annual DOM composition" are imprecise. We

revised this term to "annual average DOM composition" to better convey our intended meaning.

Additionally, we carefully revised the manuscript in multiple places to clarify our goals and relate to these in a more straightforward way in the results and discussion sections. To better convey our conceptual strategy and the study design we created a conceptual graph, which we now introduce as the first figure. This graph illustrates our study design and summarises the three main responses analysed as components of "DOM regimes".

Last, we also opted for a change in our general storytelling: The observed different responses in Atlantic and Mediterranean rivers with regard to DOM regimes following flow alteration were largely owed to the differences in how DOM regimes behave in natural flow, i.e. the starting conditions of DOM behaviour used for statistical comparisons. Instead of focusing on different responses in Atlantic and Mediterranean rivers, we now focus on the consistencies in the reaction to dams that appear despite the naturally different starting conditions.

We believe these measures improved the manuscript's readability and addressed the raised concerns regarding the difficulty in grasping the study's goals and findings.

Specific comments

L30-32: "This highly reactive fraction…" a reference is needed.

- Here, we cited Kaplan et al., 2008 and Hansen et al., 2016 to support the sentence.

L34 but also temporally (Catalán et al., 2016)… not an adequate citation in that context, because that work really looks at spatial differences of a time-reated property

- We rephrased this text section to better reflect the linkage of variations of DOM in time and space in rivers with appropriate citations (Line 36).

L53 This rather general model of a DOM regime´s reaction to damming needs fine-tuning… quite a colloquial language for the central part of the study motivation

- We changed the language here (Line 57).

L55 inflowing DOM concentration: not really the concentration but the amount

- We corrected this (Line 55)

L56 I don't agree that "all" these biotic factors are "associated" with the natural flow regime

- We changed this to 'most' are 'associated' (Line 58)

L 58, the term "compositional turnover of DOM" needs to be clearly defined, see above general comment.

- We changed this wording and defined what we mean by it here in Line 61 where it first appears.

L63 two naturally defined hydrological classes,.. this is the first appearance outside of the Abstract and the relevance of this concept demands appropriate introduction on first appearance

- We clarified what we mean by natural regimes and opted to not use the "hydrological class" term for the sake of limiting the diversity of vocabulary. Instead we now sometimes use the phrase "reference flow regime(s)" when referring to a hypothetical pre-damming flow regime of a river.

L65 We expect the effect of flow regime alterations on the DOM regime to depend on certain characteristics of the natural flow regime. … this is an unintuitive research goal, what "characteristics" could this be?

- We apologise for the confusion, this was not well formulated. In fact, we here merely wanted to suggest that the effects of flow regime alterations on the DOM regime will depend on the initially unaltered flow/DOM regimes. Natural flow regimes (and likely also associated DOM regimes) are quite diverse. And even if flow alterations are also diverse, any alteration of the DOM regime still happens from a certain baseline dictated by the natural flow regime. We reformulated this part of our research objectives (Lines 67-70) and supported our reasoning with a conceptual figure (Fig 1) laying out our study design and hypothesis framework.

L 161-163, the authors state that the sampling dates to the centroid of a river serves as a measure of temporal turnover of DOM and it is computed as a dispersion. There is not a clear explanation of what this dispersion precicely means and how it is derived. More explanation would be useful.

- We replaced the phrase "temporal turnover" by the less ambiguous phrase "temporal shifts in composition" and added further explanations for the multivariate measure of variation (i.e. "dispersion") in lines 182-184. A summary of our used statistical measures is also included in Fig.1 to visually demonstrate how these measures are tied to temporal and spatial variation.

L259, what are the "temporal turnover indicators"? These indicators are not explicitly defined.

- We changed this term to "compositional shift indicators" here in Line 286 and explained in Methods (Lines 156-157) that temporal coefficients of variation (CVs) are used as temporal shift indicators of DOM composition at the annual timescale.

L309 blurry but more encompassing… not sure I understand what you mean here

- We cleared the sentence indicating that the 3rd model has a lower resolution (due to it combining all the PC axes) but might be more comprehensive (Lines 344).

L 361, "hydropeaking" may need definition or referencing

- We added a reference to Almeida (2020) as an example for literature defining hydropeaking.

L477-479, sentence quite long, consider dividing

- We shortened this sentence.

Fig 2 add aM irrigation

- Thank you for noticing, we added this to the legend.

Table 4 sample n and frequency may be a useful information here

- We added sample numbers beside the regimes and indicated the frequency in the Figure caption.

Figure 7 flow properties: these should be introduced at one point earlier in the text. Maybe revise the image altogether because it is hard to read and unexpectedly complex for this stage of the manuscript. Why not for example instead of grouping by category, sort by influence, or withdraw from showing *all* influences and select the most significant/important ones. I believe this would increase the interaction with the information massively.

Thank you for your comments. Indeed this figure is very complex and we have tried many alternatives for the sake of saving space and in order to increase clarity. We believe dividing it even more would create more confusion than providing clearance at this point. The figure already shows only a subset of all the indices used in the model, selected according to VIP values. Also, we wish to keep the aspect of comparability among the 3 models built for the three response variables, as this makes the plot both informative as well as space-efficient. The current plot version also nicely shows that there is a trend throughout the year in flow magnitudes (positive correlation in winter, negative in summer) as well as in other indices such as minimum-maximum daily to monthly events and other contrasting indices of high water vs low.

To increase interaction with the information, we grouped the indices of the same group in sections. This way, we hope such trends throughout the year can be seen as a set, conveying one message.

494 subscript $CO_2$
- We corrected this.

Citations of the Xenopoulos review show up several times. Maybe it is useful to cite original study in some cases?

- We updated these references to the original study in line 445 and line 450.

**Citation**: https://doi.org/10.5194/egusphere-2024-2772-RC1

**Reviewer #2, Ji-Hyung Park (AE):**

Unexpectedly, two appointed referees could not provide review reports even after the extended due date. Therefore, Associate Editor provides another required report to expedite the delayed review process. – Ji-Hyung Park

General comments

This study investigates the effects of natural and altered flow regimes on DOM concentration and composition in several rivers across Northern Spain. It is a well-designed study employing a combination of analytical tools and advanced statistical approaches. The key findings on the complex linkages between flow regimes and DOM are novel and intriguing. Although the manuscript is generally well written, it also requires both clarifications of several uncertainties and improvements for a more focused discussion as follows:

We thank Ji-Hyung Park for their positive evaluation of our work.

1. Hypotheses and key themes: This study examines the effects of altered flow regimes under two different natural settings. As the authors stated ("We hypothesize that DOM properties respond to both natural flow regimes and flow alterations. We expect the effect of flow regime alterations on the DOM regime to depend on certain characteristics of the natural flow regime."), the concentration and composition of DOM would respond to changes in natural flow characteristics without any anthropogenic perturbations to flow regimes. Therefore, the quite general hypothesis needs to be more articulated with regard to which specific DOM characteristics would respond to which specific characteristics of flow regimes (both natural and anthropogenic). Furthermore, the dams in the studied region appear located in headwaters or upper reaches. Given the fact that flow regime changes vary with dam types and locations, dam effects, as a key theme of this study, need to be explained for the prevailing dam characteristics of the study region.

Thank you for your comments. We revised our hypothesis in Line 70-73.

In this study, our primary focus was on examining DOM changes following flow alteration under two different natural flow regimes, irrespective of dam types or purposes behind the flow alteration.

While we acknowledge that specific regional climates shape flow regimes but also influence the design and operation of reservoirs, we aimed to identify overarching trends rather than focus on individual dam characteristics. However, while our study design simply refers to "dams" altering flow, we now acknowledge that dams actually serve various purposes in the way we present and discuss our results. We discussed the "dam characteristics and operations" relevant to altered flow regimes in Section 3.1, where we connected these characteristics to the water demands of the region. However, to provide greater clarity, we expanded on the common features of the dams in our study, such as their locations (e.g., predominantly in headwaters or upper reaches), the absence of hydropeaking in all but one, and their storage capacities in Line 106-108 to offer a clearer context for the observed dam effects without going into the details of a site-based analysis.

Also with regard to DOM changes, our a priori formulated hypothesis was simply that dams would have an effect on DOM and our intention was to identify mechanisms behind those changes. In this regard it is important to note that we relied on optical features of DOM, that are mostly understood as proxies for actual chemical traits and as such may just indicate certain mechanisms acting on and defining a DOM regime. We therefore refrain from defining hypotheses about how dams may change specific DOM features.

Also, our analysis largely uses descriptors of DOM regimes that are fairly derived, e.g. describing temporal shifts of DOM composition. In this light, we have decided to write both the introduction as well as the discussion leaning more towards integrative DOM regime descriptors and mechanisms that affect DOM regimes. Of course, individual DOM descriptors, e.g. a fluorescence intensity ratio, may be presented to support our reasoning. Note that Table 4, Supp Fig S2 e-f presents ample information about individual DOM features and their behaviour.

2. River classification: The authors describe as if they studied 20 different rivers, but actually Fig. 1 shows 20 river locations. Some of them, as tributaries, belong to the same river basins. Around five basins are delineated on Fig. 1. Please articulate how many river basins and their tributaries were studied; and use relevant terms to distinguish mainstems and tributaries. In the same context, the river classifications in L 78-81 need revision. Some of the river sites belonging to two different systems (Atlantic and Mediterranean) are shown shoulder to shoulder on Fig. 1, making me wonder whether these sites really belong to two different "climatic regions". For instance, look at site Arlanzon. From the names, I expected that these rivers might discharge into either the Atlantic Ocean or the Mediterranean Sea. In my view, the use of these names needs to be confined to indicate something like "Atlantic-type flow regime" after defining it at its first use.

Thank you for pointing this out. The layout and terminology we used indeed reflect some complexities, partly due to reliance on a previous study which put constraints on our study design. In the original study by Peñas et al. (2020), Spanish river systems were classified based solely on hydrological indices, without considering their climatic regions. Subsequently, we assigned the names "Atlantic" and "Mediterranean" to the hydrological groups based on their predominant flow regime characteristics—Mediterranean rivers typically experience long, dry summers, while Atlantic rivers exhibit shorter summers with relatively temperate flow regimes year-round. We recognise that this naming convention might not be intuitive at first glance, as it is based on hydrological patterns rather than geographic location or basin.

To address your concerns, we clarified in the manuscript that these classifications represent "Atlantic-type" and "Mediterranean-type" flow regimes (line 88), and they are not indicative of where the rivers discharge or the climate zone their basins belong to, in line 86-90.

Regarding your observation about tributaries, it is correct that some of the studied "rivers" are tributaries within the same river basin. However, we checked the spatial independence of the studied rivers and found that all except Aguilar are spatially independent. We have indicated so in line 87. We have revised the methods section and added the number of basins in line 86 and clarified that we use the term "river" when referring to the sampling sites for simplicity in line 91. We have also added the stream order in Supplementary Table S1.

3. Data interpretation: According to Table 4, DOC concentrations are significantly different only between nA and Am. Associated descriptions in Abstract and Results (L 226-230) need to be consistent with the presented results. Although the means were not significantly different, inter-regime differences distinct for some of the river sites (Fig. 4) warrant

descriptions in L 226-230, particularly in relation to (any common) specific hydrologic characteristics of these sites. Statistical analyses indicated the dominant role of high-flow periods (L 311-313) in explaining variations in DOM composition. Can this be related to flow-controlled variations in DOC conc.? Regarding the lack of dam effects on DOM composition indices, please refer to specific comments below.

Thank you for your comments. You are correct that the DOC concentrations were significantly different only between nA and aM. However, while we conducted pairwise statistical tests between all possible pairs, we only adjusted P-values given a limited a priori planned testing design that excluded the pairs nA/aM and aA/nM. This decision was intentional, as our primary focus was to compare rivers within the same regime or to compare the two natural regimes to emphasize their distinct characteristics. Unfortunately, the differences between natural and altered were not significant for both A and M rivers, yet trends were emerging, letting us suspect an issue of statistical power. We took care that these non-significant results are clearly reported as "trends" only.

Regarding your suggestion about inter-regime differences emerging from some sites, we have revised the text in lines 254-259 to incorporate comparisons of rivers within aM that show extremely high DOC values. Upon inspection, we couldn't find a common driver for such behaviour but we think the high storage capacity of Ebro and Duero combined with their reservoir release and filling times may be the driving force behind the distinctly high DOC of these rivers. We have also added a comparison of the discharge values and DOC extremes of these two rivers showing an opposite trend where high DOC values are observed at low discharge in one while the trend is opposite in the other.

Specific comments

- Title: Do you mean that…two distinct hydrological regimes "of" Northern Spain?

 - Changed.

- Line (L) 2: What have altered flow regimes?

 - Many rivers specifically from this region have altered flow regimes, we indicated so in the text.

- L 3 and thereafter: Please remove the comma after the subject and check grammatical errors and typos throughout the manuscript including tables and figures (for instance, superscripts such as m2 and concentration units in parentheses).

 - Thank you for noticing, we corrected such errors.

- L 6 "hydrological classes": As suggested by the first reviewer, please use terms consistently; in this case "flow regimes"? Definitions are required for the key terms including flow regimes, DOM regimes, and DOM turnover.

- We agree that these terms create confusion at various points in the manuscripts. We adapted our manuscript to clearly identify a set of terms early in the manuscript and also in the conceptual graph, and maintained usage of the same vocabulary throughout the text. Instead of "DOM turnover" we used "compositional shifts of DOM" throughout the text to avoid ambiguity. We tried to avoid the term "hydrological class" and opted for "flow regime" wherever possible and included explanations when not.

- L 8: Please do not divide this relatively short abstract into two paragraphs.

- We combined these two paragraphs.

- L 12: Was this turnover rate calculated per unit carbon? Higher DOC concentrations might have led to higher turnover rates, so this needs to be clarified.

- The here previously used word "turnover" referred to temporal changes in DOM composition. We abandoned the usage of "turnover" for this purpose and instead refer to "shifts in DOM composition", as also indicated in the conceptual graph. With this in mind, we hope we made it clear that the paper is not looking into "carbon turnover rates" but into temporal shifts in carbon composition, i.e. DOM regimes.

- L 14: What are "unusual DOM behavior"?

- We changed this to "deviations from the natural DOM behaviour".

- L 19: Please specify "these" in this complex sentence.

- We changed "these" to "natural flows".

- L 53 "DOM regime´s reaction to damming needs fine-tuning": Please rewrite this vague sentence.

- We removed this statement.

- L 58 "compositional turnover": Compositional changes, or variations?

- Here we refer to the temporal changes in DOM composition, we updated this expression accordingly. We abandoned usage of the word "turnover" to describe compositional changes throughout the manuscript.

- L 93-94: Please provide more details about the hydroclimatic conditions for these six samplings. Later in Results and Discussion, sampling conditions need to be related to flow regimes and DOM characteristics. It would be helpful if sampling timings are indicated on Fig. 2.

- We indicated the sampling timings in Fig.2 with grey shades and in the text in line 109. We judged the idea of indicating detailed hydroclimatic conditions for each site per sampling occasion to require too much space in the text while simultaneously not being very

informative as our sites had varying conditions from dry to flooded even within the same sampling campaign. However, to give more insight into site-specific hydroclimatic conditions, we provided discharge data in the supplementary Table S1 per sampling occasion.

- L 99-105: Given the importance of DOC quantification and optical characterization in this study, please provide some key QA/QC measures, including blanks, verification standards, references,,,

      - We provided more details about DOC and optical characterisation of DOM in lines 116-121

- L 218-211: Please describe in more detail about the meaning of and how to read out the magnitude differences. It is very difficult to follow!

      - We added here a short description of what the short duration and seasonal low and high magnitude events are in line 241.

- L 226: It would help understand the temporal variations in DOC conc. if variations in hydrologic conditions among the six samplings were described briefly.

      - Per your recommendation, we added representative sampling dates to Fig.2 and provided the discharge measurements on the sampling dates in Supp. Table S1. We hope this will help to understand these DOC variations as well.

- L 266-267: Interesting approach! The four regimes overlap considerably on Fig. 5a. Do some outlying sites also exhibit high DOC or any common outstanding hydrologic characteristics?

      - Indeed, we have looked at whether these extreme sites/occasions coincide with high DOC values but could not find any relation.

      We have checked these outlier sites and found no abnormally high DOC concentrations or extreme discharge values for those specific sites. Two of these extreme sites show normal DOC concentrations at normal discharges measured on different sampling dates. Another one has, contrary to the suggestion, very low DOC at low discharge. We added a statement that we couldn't find a common DOC or discharge trend between the outlying measurements in lines 256-259.

- L 307: Again this 'turnover' is an undefined, ambiguous term.

      - We updated this term to "shifts" as stated above.

- L 324-337: The dams studied in this study appear located in headwater streams. Dams in lowland or high-order streams might be quite different in terms of flow regimes. Please consider this stream-order effect.

- Indeed all dams are on 2nd, 3rd or 4th order streams. We included stream orders in Suppl. Table S1. We also emphasized that our altered flow regime sites were all impacted by headwater dams in line 100.

- Discussion is very long and repeating some results descriptions, but limited in citing other relevant studies to put the implications of the key findings in a wider regional or global context. Please consider removing repeated descriptions while focusing discussions on key implications and comparisons with previous studies.

- The discussion was extensively rewritten. We removed the repetitions, especially in section 4.1, and added more comparisons with the literature, especially in section 4.4.

- L 331 (and other places): Please use two instead of 2.

- We updated this in the manuscript.

- L 351: This very long paragraph can be split here.

- We divided this paragraph in 2.

- L 373-375: Please don't repeat the descriptions already mentioned in Results.

- We updated this sentence to not repeat results (see lines 409-414).

- L 382: not "increase" but "increasing trend".

- We update this sentence to "increasing trend" in line 422.

- L 387: How can "hydrological averaging" can increase DOC concentrations? DOC flushing mechanisms explained by Raymond et al. assume higher DOC concentrations at terrestrial sources. Increased DOC conc. in stream downstream of reservoirs should then indicate additional supplies from autochthonous sources.

- We suggest that hydrological averaging by dams that hold back flood water with high DOC may lead to an increased and stable mean concentration of terrigenous DOC downstream of the dam. In contrast, with limited sampling happening only several times a year it is very unlikely that, in a natural flow regime, such short-term high DOC floods can be captured. The result is - merely considering terrigenous DOC mobilized by floods and sampling effects, higher DOC downstream of dams. Indeed, this requires a range of conditions, which we did not detail well enough, namely fairly high terrigenous DOM loads mobilized by floods and their rather bio- and photo-resistant behaviour. Also, any such terrigenous DOM must still be understood in concert with autochthonous DOM produced in the reservoir. We improved the description of our idea of "dams flattening a DOM/DOC pulse" in the revised manuscript and described the conditions upon which it may hold. We have added a more detailed explanation in line 425-431 and touched upon it further in the conclusion (per your comment L487).

- L 388-391 & 398-410: Given the significance of dam effects as a key theme, a more in-depth discussion, including comparison with previous studies on DOM characterization in other dam types (not citing only one review paper), is required to explain the observed patterns in aA and aM.

- Thank you for your suggestion. We have included a comparison in lines 414-418 on reservoir effects on DOC concentration and also a similar comparison in lines 443-450 on reservoir effects on DOM characterization. We have also added a couple more references to the general explanation of dam effects on DOM composition in lines 452-457.

- L 392-397: Does the lack of statistical differences in DOM composition indices simply translate into no dam effects on DOM? As suggested before, a more detailed look at large within-regime variations is required to figure out some key characteristics driving the link between dam-induced alterations in hydrology and DOM. For example, you are not addressing the trophic status of your reservoirs, which may be quite different from eutrophic reservoirs.

- Indeed, an eventually observed DOM regime downstream of a dam may be driven by a large range of potential influences associated with specific dam features, e.g., location in the continuum or catchment size, reservoir size and residence time, trophic status, and operation regime. Our study design lumps together dams and reservoirs of various types, thereby achieving reasonable but admittedly not excessive statistical power with a sample size of 4-6 per treatment. To identify how specific dam factors may influence DOM would have required a different study design, likely resulting in higher effort to ensure statistical power. Despite our study design targeting general "dam" effects, data may still be looked at to further explore within-regime variation as suggested. We have modestly improved this aspect in our revised manuscript by including more information on the trophic status and residence time of water in reservoirs, detailing our efforts in this respect in results and discussion, yet refraining from putting more focus on this given low statistical confidence in eventual findings. We also simply lack more detailed information about dam operation and prefer to refrain from further speculative interpretation of our data that is not oriented along the principal design of the study

We would like to point out that trophic status of reservoirs (Table 1) was in general biased towards oligotrophic conditions and of limited help in explaining within-regime variation but used as supporting information on within reservoir DOM production (see line 524). Storage capacity and associated residence time may explain some of the observed among-river differences and we specifically discussed this in section 4.3. Exploring dam purposes was a bit more fruitful and we point to lines 389-396 for our results in this regard.

Notably, despite high within-regime variation and limited statistical power given our design, our analysis beyond the analysis of "average DOM composition" and focusing on temporal variation of DOM composition, i.e. a "DOM regime", identified a few common aspects of DOM regimes following flow alteration by dams. In fact, we found dams to diversify flow regimes in downstream rivers but associated DOM regimes rather showed signs of spatial and temporal homogenization. Notably, these findings appeared fairly robust

across natural flow regimes, even in the light of natural variation and variation associated with dam heterogeneity.

- L 408: Yes, but which "factors related to dam nature and operation" are critical for your rivers?

- Please see our answer to the previous comment. Besides generally alluding to a range of dam-specific features that may influence DOM regimes, we have tried to look into common dam properties that might be driving the DOM composition response to flow alteration by dams.

- L 413-414: Yes, temporality is critical. However, the following descriptions only focus on the statistically analyzed means. As suggested, a brief discussion can be added here to evaluate how hydrologic conditions preceding the six samplings might have (differentially) affected the observed patterns in the four systems.

- In addition to indicating the sampling dates in Fig 2 as you've suggested, we added discharge data on the sampling day for each site in the SI to give a clear idea on sampling day conditions per site. We refrained from adding a limited discussion of how the observed hydrograph of particular rivers may be associated with the observed DOM regime here but did expand on it along with other flow properties in the previous sections (line 488-496…).

- L 455-: Please provide some data or literature information to link these trophic changes to autochthonous DOM production.

- Thank you for noticing, here, we cited three references, Imtiyazi 2024, Ulseth and Hall 2015; Maavara et al. 2020, regarding reservoirs and residence times and reservoirs and nutrients in lines 519-520.

- Section 4.4: The title is not in agreement with the text descriptions. This short paragraph is not redundant? Otherwise, provide a more proper discussion.

- This section closes our discussion with a last point looking at among-river variation in flow regimes. We regard spatial variation of DOM regimes across rivers as a separate response and hopefully made that clearer now right from the start (Fig1). We altered and integrated this section into the previous section given its short length.

- L 487 "longer residence times average out the naturally high turnover of DOM composition, and send relatively invariable DOM further downstream.": This conclusion cannot explain the observed increasing DOC trend and is not in line with the following implication "increased metabolism of terrigenous DOM and increasingly higher CO2 emissions in rivers downstream of dams". From my own experiences in highly eutrophic, impounded systems, CO2 emissions generally increase with the increasing supply of labile DOM (please refer to this review: https://doi.org/10.1016/j.watres.2022.119362).

- Please see our above explanation of how hydrological averaging by transiently storing flood waters in reservoirs may lead to apparently higher terrigenous DOM downstream of dams given the limited sampling effort. We improved our explanation of this potential mechanism and the conditions under which it may hold. Our idea of increased metabolism of this DOM downstream of dams is based on the conception of an improved match of temporally invariant DOM chemistry with microbial capabilities. This is a rather speculative idea based on the idea that microbial functions must fit chemical DOM traits to allow uptake and respiration or conversion into biomass. It is a mechanism decisively different from the view that a principal DOM lability entirely controls its usage by microbes (which to some extent is certainly true). We brought up this speculative idea late in the manuscript (lines 513-517, 556) to perhaps spur future studies into an interesting implication of dam effects on DOM regimes. Potentially, DOM downstream of dams could be biased towards labile material produced in the reservoir (especially when these are of higher trophic status), while simultaneously being fairly stable in terms of composition (this point potentially being valid for autochthonous as well as allochthonous components) - both effects could increase DOM metabolism, but by different mechanisms. There is plenty of room for further studies here and we pointed to one possible avenue.

- Fig. 1: Given the significance of impoundments in distinguishing flow regime types, I wondered if dam locations could be indicated within each altered-regime subcatchment.

- We included dam locations for rivers with altered flow regimes in our map.

---

## Author Response (AR2)

**Anonymous Reviewer:**

- Figures 4-7 would profit from a slightly more homogenious visualisation of axis labels, maybe worth looking at (units in parantheses, "PC1 (unitless)", magnitude of flow events).

**Response**: We appreciate the reviewer's feedback. We have revised the axis labels in Figures 4, 5, and 7 to ensure consistency in the presentation of units. For Figure 6, we have retained the original axis title with the explained variation in parentheses, as this format is standard practice for PCA plots.